# Monopoles in Dirac spin liquids and their symmetries from instanton calculus

G. Shankar[1] and Joseph Maciejko[1,2]

**1** Department of Physics, University of Alberta, Edmonton, Alberta T6G 2E1, Canada
**2** Theoretical Physics Institute, University of Alberta, Edmonton, Alberta T6G 2E1, Canada

October 25, 2023

## Abstract

The Dirac spin liquid (DSL) is a two-dimensional (2D) fractionalized Mott insulator featuring massless Dirac spinon excitations coupled to a compact $U(1)$ gauge field, which allows for flux-tunneling instanton events described by magnetic monopoles in (2+1)D Euclidean spacetime. The state-operator correspondence of conformal field theory has been used recently to define associated monopole operators and determine their quantum numbers, which encode the microscopic symmetries of conventional ordered phases proximate to the DSL. In this work, we utilize semiclassical instanton methods not relying on conformal invariance to construct monopole operators directly in (2+1)D spacetime as instanton-induced 't Hooft vertices, i.e., fermion-number-violating effective interactions originating from zero modes of the Euclidean Dirac operator in an instanton background. In the presence of a flavor-adjoint fermion mass, resummation of the instanton gas is shown to select the correct monopole to be proliferated, in accordance with predictions of the state-operator correspondence. We also show that our instanton-based approach is able to determine monopole quantum numbers on bipartite lattices.

# 1   Introduction

A quantum spin liquid is a quintessential example of a fractionalized phase in strongly corre-lated systems, whose low-energy description is best afforded by a deconfined gauge theory [1]. The parton construction is a systematic approach to derive such a description [2–4]. In such an approach, the lattice spins are rewritten as a composite of fermions or bosons (partons) glued together by a confining gauge field. While these partons remain confined in conventional phases, a quantum spin liquid is characterized by their deconfinement at low energy.

Of the various spin liquids that have been proposed, the Dirac spin liquid (DSL) is of special renown for its candidate role as a "parent state" for several competing orders in two spatial dimensions (2D) on various lattice geometries [5–11]. As known and reviewed below, a low-energy description of the DSL state is afforded by compact quantum electrodynamics in three spacetime dimensions (CQED$_3$) with $N_f = 4$ flavors of massless Dirac fermions. This theory is strongly coupled in the infrared and is expected to flow, at least for sufficiently large $N_f$, to an interacting conformal field theory (CFT) with an emergent $SU(N_f)$ flavor symmetry, at which one observes power-law correlations in order parameters for several microscopic competing orders [5, 8, 12].

In this story, the first question to be asked concerns the *stability* of the DSL. Are there rele-vant operators in this CFT with the same microscopic lattice symmetries as the DSL? Fermion bilinears are of course relevant, but always violate microscopic symmetries [5, 8–10]. Of special concern are monopole operators in CQED$_3$ [13–16], which have their origin in the compactness of the emergent gauge field that results from the parton construction on the lat-tice. At least for sufficiently large $N_f$, all monopole operators are irrelevant [12, 16–18] and CQED$_3$ remains in a deconfined phase, thus guaranteeing stability of the DSL. In contrast, the fate of the DSL for small $N_f$, including the value of interest $N_f = 4$, is murkier. The is-sue is the possible renewed relevancy of monopoles, in which case one then has to determine if there are monopoles with the same symmetries as the microscopic realization of the DSL on a given lattice. Correctly determining how monopole operators transform under lattice symmetries (i.e., their "quantum numbers") has been the subject of a longstanding theoreti-cal program [8–10, 19–22]. To be specific, as monopole operators in CQED$_3$ are dressed by fermion zero modes [16, 23, 24], their transformation under lattice symmetries has two contri-butions: from the zero modes themselves, and from a $U(1)_{\text{top}}$ phase shift of the bare monopole interpreted as a Berry phase obtained on dragging the monopole through a Dirac sea. (Here $U(1)_{\text{top}}$ denotes the $U(1)$ topological symmetry of planar $U(1)$ gauge theories, whose global charge is the total magnetic flux.) The latter Berry phase has been difficult to compute, and a general framework to do so has only recently emerged in two works by Song *et al.* [9, 10]. Their conclusions indicate that, on realizations of the DSL on bipartite lattices, there always exist monopoles that transform trivially under all lattice symmetries of the state. The relevancy of such monopoles will then destabilize the DSL, and a transition into one of the proximate

competing orders is then expected.

The second part of the DSL story is then determining the various competing orders for a given microscopic realization of a DSL [5, 8–10, 22, 25, 26]. The immediately available "order parameters" in the continuum field theory are the gauge-invariant fermion bilinears $\bar{\psi} t^a \psi$, where $\psi$ is a spinor in the fundamental representation of the $SU(4)$ flavor symmetry group and $t^a \in \mathfrak{su}(4)$. However, the spontaneous generation of an expectation value for such a fermion bilinear is not enough to drive the DSL into the corresponding ordered phase, for the fermions are still deconfined. To obtain phases with conventional long-range order, one further requires a mechanism by which the gauge charges confine. This is assumed to be due to monopole proliferation in the gauge theory, whose consideration we are again led to. The state-operator correspondence allows one to classify all monopole operators by their scaling dimension [16, 18, 27–37]. Combined with the methods developed in Refs. [9, 10] to compute the quantum numbers of the monopoles, one can determine the correct monopoles to add to the Lagrangian. As argued in those references, the transition from the DSL into a proximate conventionally ordered phase then consists of a two-step process in which a fermion bilinear is first spontaneously generated, due for instance to a sufficiently strong symmetry-allowed four-fermion interaction [38], followed by the proliferation of the relevant monopoles to drive confinement. In certain cases, the fermion bilinear does not encode all the broken symmetries of a given microscopic order, and monopole proliferation is responsible for breaking the remaining symmetries.

To construct these monopole operators, Ref. [16] utilized the conformal invariance of massless CQED$_3$ at large $N_f$ and defined monopole operators as states in the large-$N_f$ CFT in a background flux on $S^2 \times \mathbb{R}$. In this paper, we use the definition of monopole operators as instanton defects in the path integral [13–15, 39] to explicitly reconstruct these directly on $\mathbb{R}^3$ as terms in an effective Lagrangian, in the specific context of a DSL. Moreover, our construction is not reliant on conformal symmetry. Indeed, we specifically focus on the dynamics of confinement once a fermion mass $\bar{\psi} t^a \psi$ is added to the DSL Lagrangian. We find that such an "adjoint mass" results in the existence of Euclidean zero modes (of the 3D massive Dirac operator) bound to instantons, distinct from the zero-*energy* modes that appear in the massless limit. Resumming the instanton gas results in the generation of an instanton-induced term in the effective Lagrangian dubbed the 't Hooft vertex [24, 40–44], which in this case turns out to be equivalent to the zero mode-dressed monopole operator found in the CFT approach. For ordered phases with (broken) symmetries fully captured by a fermion mass, we show that requiring the associated 't Hooft vertex to satisfy the same symmetries can be sufficient to compute monopole quantum numbers under microscopic symmetries. As observed in Refs. [9, 10], the DSL on square and honeycomb lattices possesses such proximate orders, in contrast to non-bipartite lattices.

The rest of the paper is structured as follows. After a review of the parton construction of the DSL in Sec. 2, we organize the effects of monopoles in the path integral as an instanton-gas sum in Sec. 3.1, where it is also shown that such instanton-bound zero modes kill the path integral. The physical meaning of these Euclidean zero modes, and their relation to zero-energy modes found in previous constructions in the literature, are discussed in Sec. 3.2. Section 3.3 discusses the technical computation of the 't Hooft vertex by resumming the instanton gas. This 't Hooft vertex is rewritten by introducing "zero-mode operators" in Sec. 4, which reveals the relation to monopole operators constructed in the CFT approach. After discussing the continuum symmetries of the instanton-induced monopole operators, we comment in Sec. 5 on their quantum numbers under lattice symmetries for bipartite lattices, and finally conclude in Sec. 6.

## 2 Review of Dirac spin liquids

For concreteness, we consider the spin-1/2 antiferromagnetic Heisenberg model,

$$H = \sum_{ij} \mathcal{J}_{ij} \boldsymbol{S}_i \cdot \boldsymbol{S}_j, \tag{1}$$

on an arbitrary planar lattice, although one really has in mind an equivalence class of lattice models differing by symmetry-allowed terms. To obtain spin-liquid states, one typically begins with a parton representation [4],

$$\boldsymbol{S}_i = \frac{1}{2} \sum_{\alpha,\beta=\uparrow,\downarrow} c^\dagger_{i\alpha} \boldsymbol{\sigma}_{\alpha\beta} c_{i\beta}, \tag{2}$$

where $c_{i\alpha}$ are fermions of spin-1/2 and $\boldsymbol{\sigma} = (\sigma_x, \sigma_y, \sigma_z)$ is the Pauli vector. Since the local spin-1/2 Hilbert space is only two-dimensional, the parton representation introduces a gauge redundancy, and one must project out unphysical states using the single-occupancy constraint:

$$\sum_\alpha c^\dagger_{i\alpha} c_{i\alpha} = 1. \tag{3}$$

The gauge group can be seen to be $SU(2)$, for a local $SU(2)$ rotation of the Nambu spinor $(f_{i\uparrow} \quad f^\dagger_{i\downarrow})$ leaves the spin operator (2) invariant.

The Heisenberg model then becomes a quartic interaction of fermions, which can be exactly decoupled inside a path integral using Hubbard-Stratonovich (HS) fields, as a prelude to mean-field theory. Motivated by a search for translationally and rotationally invariant spin liquids[1], the most general decoupling consistent with these requirements results in a Lagrangian (assuming sums over repeated spin indices):

$$L = \sum_i c^\dagger_{i\alpha} \partial_\tau c_{i\alpha} - \sum_{ij} \frac{\mathcal{J}_{ij}}{4} (c^\dagger_{i\alpha} z_{ij} c_{j\alpha} + \text{h.c.}) - \sum_{ij} \frac{\mathcal{J}_{ij}}{4} (\epsilon_{\alpha\beta} c^\dagger_{i\alpha} w_{ij} c^\dagger_{j\beta} + \text{h.c.})$$
$$+ \sum_{ij} \frac{\mathcal{J}_{ij}}{4} (|z_{ij}|^2 + |w_{ij}|^2) - i a_0 (c^\dagger_{i\alpha} c_{i\alpha} - 1). \tag{4}$$

Here, $a_0(\tau)$ is a Lagrange multiplier field that imposes the half-filling constraint, and $z_{ij}$ and $w_{ij}$ are complex-valued HS link fields. The saddles of $z_{ij}$ and $w_{ij}$ are respectively at $c^\dagger_{i\alpha} c_{j\alpha}$ and $\epsilon_{\alpha\beta} c^\dagger_{i\alpha} c^\dagger_{j\beta}$, so mean-field ansätze for $z_{ij}$ and $w_{ij}$ are equivalent to condensing those fermion bilinears. Introducing the Nambu variables,

$$\psi_i = \begin{pmatrix} c_{i\uparrow} \\ c^\dagger_{i\downarrow} \end{pmatrix}, \qquad T_{ij} = \begin{pmatrix} z_{ij} & w_{ij} \\ w^\dagger_{ij} & -z^\dagger_{ij} \end{pmatrix}, \tag{5}$$

and Pauli matrices $\tau^l$, $l = 1, 2, 3$ that act in this Nambu space, and relabeling $a_0 \to a_0^3$, the Lagrangian can be rewritten as:

$$L = \sum_i \psi^\dagger_i (\partial_\tau - i a_0^l \tau^l) \psi_i - \sum_{ij} \frac{\mathcal{J}_{ij}}{4} (\psi^\dagger_i T_{ij} \psi_j + \text{h.c.}) + \sum_{ij} \frac{\mathcal{J}_{ij}}{8} \text{tr}\, T^\dagger_{ij} T_{ij}, \tag{6}$$

---

[1] While breaking spin-rotation symmetry does not preclude a spin-liquid ground state [45], the Dirac spin liquid is a state that preserves this symmetry.

where the half-filling constraint is redundantly imposed using two more Lagrange multipliers, $a_0^1$ and $a_0^2$, to produce the temporal component $a_0 \equiv a_0^l \tau^l$ of an $\mathfrak{su}(2)$ gauge field. Indeed, the Lagrangian is now independent under an $SU(2)$ gauge transformation:

$$
\begin{aligned}
a_0(i) &\to \Omega_i (a_0 + i\partial_\tau)\Omega_i^\dagger, \\
T_{ij} &\to \Omega_i T_{ij} \Omega_j^\dagger, \\
\psi_i &\to \Omega_i \psi_i.
\end{aligned}
\tag{7}
$$

The Lagrangian (6) is an exact representation of the spin-1/2 Heisenberg model on an arbitrary lattice, and describes a lattice $SU(2)$ gauge theory at infinite gauge coupling (i.e., with no dynamics for the gauge fields), but with the group elements $U_{ij}$ on every link being arbitrary complex matrices instead of $SU(2)$ matrices. However, any complex matrix admits a polar decomposition

$$
T = \sqrt{T^\dagger T}\, U \equiv \rho U,
\tag{8}
$$

where $U$ is unitary, and $\rho$ is positive semi-definite and Hermitian.

At this point, one chooses a mean-field ansatz $\langle T_{jk} \rangle$ that renders the parton Hamiltonian quadratic. As $T_{jk}$ is gauge-covariant, this ansatz generically violates gauge invariance, and the mean-field Hamiltonian $H_{\mathrm{mf}}$ will not commute with the constraint $\mathrm{tr}\,\psi_i^\dagger \tau^l \psi_i$. However, some measure of gauge invariance is restored by considering fluctuations in $T_{jk}$ about its mean-field value. Of these, there are "amplitude fluctuations" in $\rho$ and "phase fluctuations" in $U$, as evident from (8). Since $\rho$ only modulates the magnitude of the hopping, it is expected that the fluctuations of qualitative importance are those of the "phase matrix" $U$. Since we are interested in the infrared fate of the system, these gauge fluctuations will have dynamics due to a renormalization of the gauge coupling to finite values under RG flow of (6). This means the hard gauge constraint (3) will be softened in the infrared to

$$
(\partial E^l)_i = \mathrm{tr}\,\psi_i^\dagger \tau^l \psi_i,
\tag{9}
$$

where the left-hand side is the lattice divergence of the electric field. It is understood that the fermions on the right-hand side are now renormalized fermions, and thus need not obey the hard constraint of the ultraviolet partons originally used in the parton construction. The mean-field Hamiltonian is then understood as written in terms of these renormalized partons, dubbed spinons.

Then writing $T_{ij} = \bar{T}_{ij}\exp(ia_{ij})$ to allow for phase fluctuations, it is intuitive from (6) that a generic mean-field value $\bar{T}$, which translates to condensing bilinears of type $c_{i\alpha}c_{j\alpha}^\dagger$ and $c_{i\uparrow}c_{j\downarrow}$, might Higgs the $\mathfrak{su}(2)$ gauge bosons down to some subgroup. A criterion given by Wen determines the infrared gauge group [2, 4, 46]. Considering all based loops on the lattice, a collinear flux (in some direction in $SU(2)$ space) of the mean-field $\bar{T}$ through all such loops results in a Higgsing of $SU(2) \to U(1)$, and generic non-collinear fluxes will break it down to $\mathbb{Z}_2$, completely gapping out all gauge bosons. On the other hand, a trivial $SU(2)$ flux ($\propto \mathbb{I}$) ensures all the $\mathfrak{su}(2)$ gauge bosons remain massless. We shall be specifically interested in mean-field states that Higgs $SU(2) \to U(1)$ on various lattices. Examples include the staggered flux state on the square lattice [5], or the $\pi$ flux state on the kagome lattice [6–8]. The spinons $(c_\uparrow, c_\downarrow)$ in these states have relativistic dispersions, with generically two Dirac nodes ($\alpha = \pm$) in the bandstructure. A linearized description at these nodes with low-energy fields $\psi_{\alpha\sigma}$, accounting for $U(1)$ gauge fluctuations, is then given by the continuum (Euclidean) Lagrangian:

$$
\mathcal{L} = \bar{\psi}(\slashed{\partial} - i\slashed{a})\psi + \frac{1}{4e^2}f^2,
\tag{10}
$$

where $f_{\mu\nu} = \partial_\mu a_\nu - \partial_\nu a_\mu$ is the field strength tensor, $\psi$ is a Dirac 2-spinor in the fundamental representation of $SU(4)$, the gamma matrices $(\gamma_1, \gamma_2, \gamma_3) = (\gamma_x, \gamma_y, \gamma_z)$ are chosen as the three

Pauli matrices, and the Dirac adjoint is $\bar{\psi} = \psi^\dagger \gamma_3$. Since the gauge coupling $e^2$ has dimensions of inverse length, the Lagrangian is expected to be strongly coupled in the infrared, flowing to an interacting conformal fixed point which we shall call the DSL fixed point. In the $1/N_f$ expansion, one can show that this fixed point becomes nearly free, characterized by $e_*^2 \propto N_f^{-1}$, so that in the limit $N_f \to \infty$, gauge fluctuations are suppressed and spinons are free [12, 47, 48]. While it is unclear if this fixed point persists as $N_f$ is lowered to the physically relevant value $N_f = 4$, conformal invariance at large $N_f$ provides an accessible window to find relevant operators that can destabilize the DSL. Of central importance are monopole operators arising from the compactness of $a$, which when proliferated act to confine spinons into gauge-neutral spins [13–15], yielding conventional phases of the parent spin system.

These monopole operators can be defined at the large-$N_f$ DSL fixed point via the state-operator correspondence in radial quantization, by considering free fermions on a sphere containing a monopole (plus fluctuations controlled by the $1/N_f$ expansion) [16]. The monopole with smallest scaling dimension corresponds to the ground state of the fermions. In a $2\pi$ flux background created by a minimal monopole, there is one zero-energy mode per flavor of relativistic fermion as required by the Atiyah–Singer index theorem. To obtain a gauge-invariant state respecting the constraint (3), half of the four zero-energy modes have to be filled. There are thus $\binom{4}{2} = 6$ monopole operators of minimal charge. If there is a symmetry-allowed relevant monopole, then the DSL is an unstable critical point separating ordered phases. If all monopoles are irrelevant, then there is no confinement and a stable DSL is obtained. However, there could be other interactions that drive symmetry-breaking by generating a fermion mass, allowing a previously disallowed monopole to then condense, causing confinement. We will now proceed to explicitly construct these monopole operators without relying on conformal invariance. As a byproduct of such a construction, we will obtain the exact monopole that proliferates for a given pattern of symmetry breaking described by the "adjoint masses":

$$M^a = m\bar{\psi} t^a \psi, \qquad t^a \in \{\sigma^i, \mu^i, \sigma^i \mu^j\}, \tag{11}$$

where $\sigma_i, \mu_i$ are Pauli matrices that act on spin and nodal indices respectively. In the CFT picture, such a mass spoils conformal invariance and splits the degeneracy between the four zero-energy modes, causing one particular combination of the six monopole operators to lower its scaling dimension compared to the rest [26]. Our construction will directly yield this monopole, and by varying the adjoint mass yields all linearly independent monopole operators.

## 3 The 't Hooft vertex

The basic idea behind our construction is to (1) formulate the instanton problem in its original Euclidean path-integral language, rather than the canonical-quantization formalism of CFT, and (2) utilize semiclassical instanton calculus [42–44] to resum a monopole-instanton gas in the presence of massive fermions [24]. We show that the existence of instanton-bound fermion zero modes (ZMs) of the *Euclidean* Dirac operator on $\mathbb{R}^3$ cause transition amplitudes to vanish unless fermion insertions can "soak up" these ZMs in the path-integral measure. This is in contrast to the massless case, where no such Euclidean ZMs exist [49]. These insertions will then "dress" the bare monopole operator that simply creates $2\pi$ flux in the gauge theory.

### 3.1 Euclidean fermion zero modes

To set up our semiclassical calculation, we decompose the emergent gauge field as:

$$a = \mathcal{A} + \delta a, \tag{12}$$

where $\mathcal{A}$ is a monopole-instanton solving the Euclidean equations of motion, and $\delta a$ describes smooth fluctuations (photons) around the instanton solution. Temporarily neglecting the coupling of fermions to photons[2], the partition function can be written as a sum over an instanton gas [24]:

$$Z = \int Da\, e^{-\frac{e^2}{2} \int d^3x\, (\partial_\mu \sigma)^2} \sum_{N=0}^{\infty} \frac{1}{N!} \prod_{k=1}^{N} \left( \int d^3 z_k \sum_{q_k \in \mathbb{Z}} e^{-q_k^2/e^2 \ell} e^{iq_k \sigma(z_k)} \int D(\bar\psi, \psi) e^{-S_f[\mathcal{A}(q_k)]} \right),$$

(13)

where $\sigma$ is the dual photon [15], $N$ is the number of monopoles in the gas, $q_k$ their charges, $z_k$ their locations (a collective coordinate), and $q_k^2/e^2\ell$ with short-distance cutoff $\ell$ (on the order of the lattice constant) the action cost for a charge-$q_k$ monopole. Finally, $S_f[\mathcal{A}(q_k)]$ is the fermion action in a *single-instanton background* specified by $(q_k, z_k)$. A dilute-gas approximation has been made in the partition above, which allows one to partition an $N$-instanton background as $\mathcal{A} = \sum_{k=1}^{N} \mathcal{A}_{(k)}$, describing $N$ well-separated boxes containing a single instanton each. Assuming a dilute gas of monopoles allows one to bring the fermion path integral inside the product in (13), and consider fermions moving in a single-instanton background instead of that of a correlated instanton liquid. This is formally accomplished by decomposing $\psi$ into fields localized in large boxes around each instanton [43], with zero overlap between boxes. This is justified in hindsight by the observation that fermion ZMs are exponentially localized on the instantons.

This paper will only be concerned with monopole operators of lowest charge $q = \pm 1$, although the computation straightforwardly generalizes to higher charges in an obvious way. The fermion path integral in Eq. (13), which we separately write as:

$$Z_f[\mathcal{A}_q] = \int D(\bar\psi, \psi) e^{-\int \bar\psi (\slashed{\partial} - i\mathcal{A}_q + mt^a)\psi},$$

(14)

evaluates to zero for a gauge-field configuration $\mathcal{A}_q$ with nonzero monopole charge $q$. This is because the Euclidean Dirac operator:

$$\mathfrak{D}_q = \slashed{\partial} - i\mathcal{A}_q + mt^a,$$

(15)

has nontrivial ZMs in an instanton background. Unlike zero-*energy* modes of the Hamiltonian [50] that are typically bound to solitons, these zero modes of $\mathfrak{D}$ are bound to instantons. The relation between energy ZMs and these Euclidean ZMs will be further elucidated in Sec. 3.2.

Explicit solutions for these ZMs are obtained in Appendix A. For a fixed mass $mt^a \in \mathfrak{su}(4)$ with $m > 0$, the normalizable ZMs of $\mathfrak{D}_\pm$ in $q = \pm 1$ backgrounds are (respectively):

$$u_+^{(i)}(r, \theta, \varphi) = \frac{\sqrt{2m}}{r} e^{-mr} \mathcal{Y}_{1,0,0}^{1/2}(\theta, \varphi) |i\rangle_a, \quad i = 2, 4;$$

(16)

$$u_-^{(i)}(r, \theta, \varphi) = \frac{\sqrt{2m}}{r} e^{-mr} \mathcal{Y}_{-1,0,0}^{1/2}(\theta, \varphi) |i\rangle_a, \quad i = 1, 3,$$

(17)

and those of $\mathfrak{D}_\pm^\dagger$ are respectively:

$$v_+^{(i)}(r, \theta, \varphi) = \frac{\sqrt{2m}}{r} e^{-mr} \mathcal{Y}_{1,0,0}^{1/2}(\theta, \varphi) |i\rangle_a, \quad i = 1, 3;$$

(18)

$$v_-^{(i)}(r, \theta, \varphi) = \frac{\sqrt{2m}}{r} e^{-mr} \mathcal{Y}_{-1,0,0}^{1/2}(\theta, \varphi) |i\rangle_a, \quad i = 2, 4,$$

(19)

---

[2]This is justified in a large-$N_f$ approximation, but one can improve the calculation by considering fluctuations around the instanton just as in Ref. [18].

where the four eigenvectors of the $\mathfrak{su}(4)$ mass are defined by:

$$t^a \, |i\rangle_a = (-1)^i \, |i\rangle_a \, , \quad i = 1, 2, 3, 4, \tag{20}$$

and $\mathcal{Y}_{q,j,M}^{j \pm 1/2}$ are monopole spinor harmonics as defined in Appendix A. As discussed in the subsequent section, gauge invariance mandates that only two of these ZMs can be filled in any fixed instanton background. It will turn out to be sufficient to consider the ZMs $u_+^{(i)}$ and $v_-^{(i)}$ of $\mathfrak{D}_+$ and $\mathfrak{D}_-^\dagger$ to obtain nearly all the results in this paper. As shown in Sec. 3.3, these lead to spontaneous fermion pair creation (in an instanton background) and annihilation events (in an anti-instanton background).

The topological guarantee of these Euclidean ZMs is provided by their relation to *energy ZMs* of a *massless* Dirac Hamiltonian in a static $2\pi q$ flux background, which are protected by an Atiyah–Singer index theorem [23]. Indeed, the Euclidean ZMs above in the limit $m \to 0$ (ignoring the normalization) have precisely the form of the energy ZMs observed in radial quantization on $S^2 \times \mathbb{R}$, on recognizing the Weyl rescaling factor $r^{-1} = \exp(-\tau)$ [16]. A nonzero mass gaps out these energy ZMs, which reincarnate as normalizable (exponentially localized) ZMs of the Euclidean Dirac operator $\mathfrak{D}$. As we show below, the physical consequence of these Euclidean ZMs is that instanton events are correlated with fermion-number violating processes.

## 3.2 Euclidean fermion zero modes in a Hamiltonian view

We first present an intuitive argument for the heretofore claimed fermion-number violating processes caused by instantons. Instead of modeling the instanton as a point source of flux spatially localized in 2D, we can distribute the $2\pi q$ flux uniformly across the area $A$ of a finite system. This is physically reasonable as a nonzero gauge coupling will supply monopoles with momentum, effectively delocalizing them. What is important is that the total flux through the system can only jump discretely through instanton events. Massive Dirac fermions under this uniform magnetic field $2\pi q / A$ are then housed in relativistic Landau levels (for each flavor),

$$\begin{aligned} E_{n\pm} &= \pm \sqrt{2\pi n |q| A^{-1} + m^2}, \qquad n \geq 1, \\ E_0 &= m \, \mathrm{sgn}(q), \end{aligned} \tag{21}$$

where $E_0$ is the "zero Landau level" obtained in the massless limit. The degeneracy of the levels is $|q|$, so that for $q = +1$ there is precisely one "zero mode" per flavor of Dirac fermion, in agreement with the Atiyah–Singer index theorem, giving a total of four modes for $N_f = 4$.

In preparation for an interpretation of instanton events, let us imagine adiabatically dialing the flux from 0 to $2\pi$. In the zero-flux limit, we simply have two bands formed by gapping a Dirac cone [Fig. 1(a)]. The single-occupancy constraint (3) ordained by the parton decomposition (2), which is equivalent to a Gauss' law constraint, mandates a half-filling of these bands (for each flavor of Dirac fermion). As a $2\pi$ flux is adiabatically turned on, the $E_{n\pm}$ levels evolve in perfect tandem out of the upper and lower bands, while the "zero mode" captures the *spectral asymmetry* of the Hamiltonian. Depending on the relative sign of $m$ and $q$, it either descends from the upper band [$\mathrm{sgn}(mq) > 0$] or ascends from the lower one [$\mathrm{sgn}(mq) < 0$]. Since we are working with $\mathfrak{su}(4)$-valued masses $m t^a$ that preserve time-reversal (TR) invariance, it follows that there are a total of 4 displaced energy "ZMs", two with energy $m$ and two with energy $-m$ [Fig. 1(b)]. Gauge invariance (i.e., the single-occupancy constraint) again requires us to fill two of these modes. The ground state is *uniquely* obtained by filling the two negative-energy modes. This is to be contrasted with the massless limit, in which all four modes are degenerate at zero energy, and there are six possible ways to fill two of them. Selecting a specific $\mathfrak{su}(4)$-mass $m t^a$ gaps the four degenerate ZMs in a TR-invariant manner, selecting precisely two of them to fill.

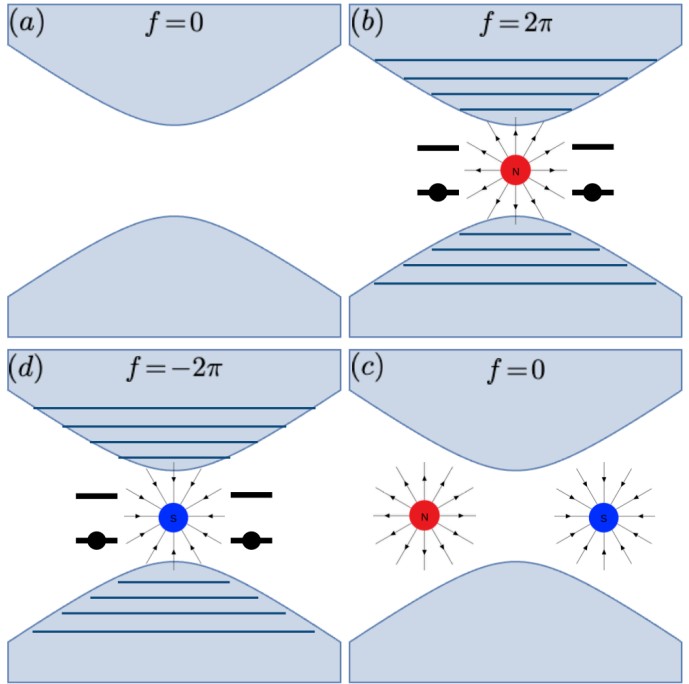

Figure 1: Clockwise from top left: Instanton (tunneling of magnetic flux $f$) events accompanied by spinon pair creation or annihilation. In a nontrivial flux sector, the Dirac bands are discretized into Landau levels with new mid-gap modes that must be half-filled to maintain gauge invariance.

Of course, an instanton is the paradigm of a non-adiabatic process. Upon a flux-tunneling event from $0 \to 2\pi$, two negative-energy modes suddenly appear in the spectrum. If these remain unfilled[3], the instanton would have caused an unphysical transition from a gauge invariant state to a non-invariant state violating the half-filling condition. The resolution is that an instanton event *must* be accompanied by fermion pair creation in the two new unfilled levels. Proceeding then in reverse from $2\pi \to 0$ flux sectors by means of an "anti-instanton", we immediately observe that anti-instantons should cause fermion pair annihilation as the two "ZMs" disappear into the lower bands [Fig. 1(c)].

These considerations lead to the conclusion that instantons cause fermion pair creation and annihilation. However, such processes must be reflected in an appropriate effective Lagrangian by means of "dressed" monopole operators of the form:

$$\mathcal{M}\bar{\psi}\Delta_+\bar{\psi}^\intercal + \mathcal{M}^\dagger\psi^\intercal\Delta_+^\dagger\psi, \tag{22}$$

where $\mathcal{M}$ is a "bare" monopole operator that creates $2\pi$ flux, $^\intercal$ denotes the transpose, and $\Delta_+$ is a vertex factor valued in $\mathfrak{su}(4)$ that will select precisely two flavors from the four $\psi_{\alpha\sigma}$ to fill the two displaced energy ZMs just discussed. Determination of this vertex factor for a specific $\mathfrak{su}(4)$ mass is one of the central goals of this work, a task that shall be taken up in the next section.

Finally, the above considerations can be equally applied to flux tunneling from $0 \to (-2\pi)$, which leads to the conclusion that anti-instantons can also create fermions [Fig. 1(d)]. This would yield a vertex contribution

$$\mathcal{M}^\dagger\bar{\psi}\Delta_-\bar{\psi}^\intercal + \mathcal{M}\psi^\intercal\Delta_-^\dagger\psi. \tag{23}$$

---

[3]It is assumed that we are at sufficiently low temperature that the leading order contribution is the filling of the two *negative* modes rather than the positive mid-gap modes that are also present. In any case, we shall see in the next section that the selection of two modes automatically falls out of the calculation.

### 3.3 Resummation of the instanton gas

In this section, the intuitive picture sketched in the previous section will be formally laid out in the path-integral framework, and the monopole operators (22-23) completely determined by a resummation of the instanton gas in the partition function (13). To do so in the presence of ZMs of the Euclidean Dirac operator, we shall use a slight variant of the technique originally devised by 't Hooft in his resolution of the $U(1)$ problem in QCD$_4$ [42–44]. More technical details can be found in Ref. [24]; see also Ref. [41] for a symmetry-based argument. An analogous calculation for $SO(N)$ gauge theory with Majorana matter was done in Ref. [40], which studied confinement transitions out of a chiral spin liquid. Readers uninterested in the technical details of the calculation can safely proceed to the next section with just the final result [Eq. (36)] in hand.

As observed in Sec. 3.1, the fermion path integral vanishes in a nontrivial instanton background, implying that only the sector with zero instanton charge contributes to the partition function itself in Eq. (13). However, sectors with nonzero charge will contribute to correlation functions that can "soak up" the ZMs (to be explained below). From the discussion in the previous section, we expect these to be correlators of the form $\langle \psi\psi \rangle$. This is best seen with mode expansions of the spinons $(\psi, \bar{\psi})$ in eigenfunctions of the self-adjoint operators $(\mathfrak{D}_+^\dagger \mathfrak{D}_+, \mathfrak{D}_+ \mathfrak{D}_+^\dagger)$ for a $q=+1$ background:

$$
\psi = u_{2+}(x-z_+)\eta_2 + u_{4+}(x-z_+)\eta_4 + \sum_i{}' w_i(x-z_+)\xi_i,
$$

$$
\bar{\psi} = \sum_i{}' \bar{w}_i(x-z_+)\bar{\xi}_i,
\tag{24}
$$

where $w_i$ are nonzero modes (indicated by the primed sums) of $\mathfrak{D}_+^\dagger \mathfrak{D}_+$, which occur in pairs with $\bar{w}_i$ of $\mathfrak{D}_+ \mathfrak{D}_+^\dagger$, and $\{\eta, \xi, \bar{\xi}\}$ are Grassmann numbers to ensure the correct Fermi statistics. The functions $u_{i+}(x-z_+)$ are the ZMs of $\mathfrak{D}_+^\dagger \mathfrak{D}_+$, localized on a charge $+1$ instanton at $z_+$, whose explicit expressions are given in Eq. (16). Only two ZMs have been included as mandated by the gauge-invariance arguments in Sec. 3.2, and the ZMs $v_+^{(i)}$ of $\mathfrak{D}_+^\dagger$ have not been "filled" by including them in the mode expansion of $\bar{\psi}$. Strictly speaking, we should sum over all possibilities by doing a separate calculation that only includes the two ZMs of $\mathfrak{D}_+^\dagger$ and not those of $\mathfrak{D}_+$. However, it will be easy to write down the result of such a calculation after our considerations below.

The functional measure can now be defined as:

$$
D(\bar{\psi}, \psi) = d\eta_2\, d\eta_4 \prod_i{}' d\bar{\xi}_i d\xi_i,
\tag{25}
$$

where the prime again denotes the exclusion of ZMs in the product. Since the ZMs $\{\eta_2, \eta_4\}$ do not appear in the Lagrangian $\bar{\psi}\mathfrak{D}_+\psi$, the Grassmann integrals over these kill the partition function. However, pair correlators of the form $\langle \psi\psi \rangle$ involve enough insertions to "soak up" the ZMs in the measure and produce a nonzero path integral. An explicit calculation, using the mode expansions (24), shows that:

$$
\left\langle \psi^a(x)\psi^b(y)^\intercal \right\rangle_+ = -K_+ u_{2+}^{[a}(x-z_+)u_{4+}^{b]}(y-z_+)^\intercal,
\tag{26}
$$

where $a, b$ are $SU(4)$ indices that have been here antisymmetrized (i.e., $v^{[a}w^{b]} \equiv v^a w^b - v^b w^a$), and $K_+$ is the fermion path integral over the nonzero modes $(\xi, \bar{\xi})$ in the instanton background. While this amplitude looks neither Lorentz nor gauge invariant at present, we reassure the reader that these issues will be addressed towards the end of the calculation.

Since the fermion ZMs are exponentially bound to the instanton with a width $m^{-1}$, this result shows that anomalous correlations also decay exponentially away from the instanton

with a length scale $m^{-1}$. This also reinforces the conclusion reached intuitively in the previous section; the fermion vacuum in the presence of $2\pi$ flux has two additional fermions compared to the one with zero flux. A transition between the two states is possible only if these two extra fermions are annihilated, and this is precisely what the $\psi\psi$ insertion achieves.

We now ask for an effective Lagrangian that reproduces such correlation functions, which will amount to resumming or "integrating out" instantons. In the pure gauge theory, it is well known that the result of such a resummation is a sine-Gordon term $\propto \cos(\sigma)$ that gaps out the dual photon $\sigma$ in the infrared [13–15]. With fermionic matter, a $2\pi$ flux is associated with two additional fermions, so we expect an effective Lagrangian to contain a term of the form $e^{i\sigma}\bar{\psi}\Delta_+\bar{\psi}^{\intercal}$. To determine $\Delta_+$, let us perturb with a generic anti-symmetrized source[4] $\psi(x)^{\intercal}J(x,y)\psi(y)$, with suppressed $\mathfrak{su}(4)$ and Lorentz indices, and perturbatively expand to $\mathcal{O}(J)$:

$$Z_f[\mathcal{A}_+, J] = \int D(\bar{\psi},\psi)e^{-\int \bar{\psi}(\slashed{D}_+ + mt^a)\psi - \int \psi^{\intercal}J\psi}$$

$$= K_+ \int d^3x \, d^3y \, u_{2+}^{\intercal}(x-z_+)J(x,y)u_{4+}(y-z_+) + \mathcal{O}(J^2), \qquad (27)$$

where the second line is obtained by using the mode expansions (24), and $K_+$ is the fermion integral over non-ZMs as in Eq. (26). Our arguments in this and previous sections have indicated that such an amplitude can be reproduced by a path integral of the form [24]:

$$I_+[J] = \int D(\bar{\psi},\psi)\, e^{-\int \bar{\psi}(\slashed{\partial} + mt^a)\psi - \int \psi^{\intercal}J\psi}$$

$$\times \int d^3x' \, d^3y' \, C_+ \, \bar{\psi}(x')\omega_+(x'-z_+)\zeta_+(y'-z_+)^{\intercal}\bar{\psi}(y')^{\intercal}, \quad (28)$$

where the vertex $\Delta_+$ has been written as a dyadic product $\omega_+\zeta_+^{\intercal}$ of vectors with possible spinor and $\mathfrak{su}(4)$ indices. We will determine $C_+$, $\omega_+$, $\zeta_+$ by demanding equality with Eq. (27) to $\mathcal{O}(J)$. Note that the fermions are no longer in a flux background in $I_+[J]$. Expanding the above integral to $\mathcal{O}(J)$ and Wick contracting gives:

$$I_+[J] = C_+ \int d^3(x,x',y,y')[G_f(x-x')\zeta_+]^{\intercal}J(x,y)[G_f(y-y')\omega_+], \qquad (29)$$

where $G_f = \langle \psi\bar{\psi}\rangle_0$ is the free fermion propagator. Comparing with Eq. (27) and demanding equality gives the vertex factors:

$$C_+ = K_+ = \det{}'\mathfrak{D}_+,$$
$$\zeta_+ = G_f^{-1}u_{2+} \approx -2\sqrt{2}\pi \mathcal{Y}_{1,0,0}^{1/2}(\theta,\varphi)|2\rangle_a,$$
$$\omega_+ = G_f^{-1}u_{4+} \approx -2\sqrt{2}\pi \mathcal{Y}_{1,0,0}^{1/2}(\theta,\varphi)|4\rangle_a, \qquad (30)$$

where we have used explicit expressions for the free propagator (Appendix B) and ZM solutions [Eq. (16)], and the approximation holds at distances $r \gg m^{-1}$ (the width of the instanton-bound ZM). One can now replace $Z_f[\mathcal{A}_+, J]$ with $I_+[J]$ in the instanton-gas sum appearing in the partition function (13).

To obtain a path integral $I_-[J] = Z_f[\mathcal{A}_-, J]$ in the anti-instanton sector, the calculation above should be repeated with ZMs of $\mathfrak{D}_-\mathfrak{D}_-^{\dagger}$ in the mode expansion of $\bar{\psi}$. One can write down

---

[4]One should strictly add $\psi^{\intercal}J\psi +$h.c., but the conjugate term cannot soak up the zero modes in the path-integral measure in the $q = +1$ sector, so we drop it to reduce clutter.

the result based solely on reflection positivity (not reality) of the Euclidean action [51,52], but since this is somewhat subtle as we shall see later, it is more prudent to just repeat the above calculation. The result is:

$$I_-[J] = C_- \int d^3(x, x', y, y') [G_f^\dagger(x - x')\zeta_-]^\mathsf{T} J(x, y) [G_f^\dagger(y - y')\omega_-], \tag{31}$$

with

$$
\begin{aligned}
C_- &= K_- = \det{}'\mathfrak{D}_-^\dagger, \\
\zeta_- &= (-\slashed{\partial} + mt^a)^{-1} u_{2-} \approx -2\sqrt{2}\pi \mathcal{Y}_{-1,0,0}^{1/2}(\theta, \varphi)|2\rangle_a, \\
\omega_- &= (-\slashed{\partial} + mt^a)^{-1} u_{4-} \approx -2\sqrt{2}\pi \mathcal{Y}_{-1,0,0}^{1/2}(\theta, \varphi)|4\rangle_a.
\end{aligned}
\tag{32}
$$

Substituting in $I_\pm[J]$ for $Z_f[\mathcal{A}_\pm, J]$ in the partition function (13), we obtain:

$$
\begin{aligned}
Z[J] = \int D\sigma\, D(\bar\psi, \psi)\, e^{-S_0 - \int (\psi^\mathsf{T} J\psi + \text{h.c.})} \sum_{N=0}^{\infty} \frac{1}{N!} \prod_{k=1}^{N} \int d^3 z_k \int d^3x\, d^3y \\
\times \left[ -K_+ e^{i\sigma(z_k)} \bar\psi(x) \mathcal{Y}_{1,0,0}^{1/2}(x - z_k) |2\rangle\langle 4| \mathcal{Y}_{1,0,0}^{1/2}(y - z_k)^\mathsf{T} \bar\psi^\mathsf{T}(y) + \text{r.c.} \right],
\end{aligned}
\tag{33}
$$

where "r.c." denotes the reflection conjugate[5], dimensionless constants have been lumped into $K$, and the free action $S_0$ is:

$$S_0 = \int d^3x \left[ \frac{e^2}{2}(\partial_\mu \sigma)^2 + \bar\psi(\slashed{\partial} + mt^a)\psi \right]. \tag{34}$$

As remarked below the mode expansions in Eq. (24), one must also sum over a transition amplitude that involves the two ZMs of $\mathfrak{D}_+^\dagger$ but not those of $\mathfrak{D}_+$. The calculations leading to Eq. (33) clearly indicate that resumming instantons with these ZMs would lead to further insertions of the kind:

$$-K_- e^{-i\sigma(z_k)} \bar\psi(x) \mathcal{Y}_{-1,0,0}^{1/2}(x - z_k) |1\rangle\langle 3| \mathcal{Y}_{-1,0,0}^{1/2}(y - z_k)^\mathsf{T} \bar\psi^\mathsf{T}(y) + \text{r.c.}, \tag{35}$$

where the ZMs (17) and (18) have been used. As predicted at the end of Sec. 3.2, this vertex corresponds to spinon-pair creation by anti-instantons. Including these terms in Eq. (33), re-exponentiating the instanton-gas sum and then setting the source $J$ to zero results in an instanton-induced contribution to the effective action: the *'t Hooft vertex*,

$$
\begin{aligned}
S_{\text{inst}}^a = &\, K_+ \int d^3z\, e^{i\sigma(z)} \left[ \int d^3x\, \bar\psi(x) \mathcal{Y}_{1,0,0}^{1/2}(x - z) \right] |2\rangle\langle 4| \left[ \int d^3y\, \mathcal{Y}_{1,0,0}^{1/2}(y - z) \bar\psi(y) \right]^\mathsf{T} \\
&+ K_- \int d^3z\, e^{-i\sigma(z)} \left[ \int d^3x\, \mathcal{Y}_{-1,0,0}^{1/2}(x - z)^\dagger \psi(x) \right]^\mathsf{T} |4\rangle\langle 2| \left[ \int d^3y\, \mathcal{Y}_{-1,0,0}^{1/2}(y - z)^\dagger \psi(y) \right] \\
&+ K_- \int d^3z\, e^{-i\sigma(z)} \left[ \int d^3x\, \bar\psi(x) \mathcal{Y}_{-1,0,0}^{1/2}(x - z) \right] |1\rangle\langle 3| \left[ \int d^3y\, \mathcal{Y}_{-1,0,0}^{1/2}(y - z) \bar\psi(y) \right]^\mathsf{T} \\
&+ K_+ \int d^3z\, e^{i\sigma(z)} \left[ \int d^3x\, \mathcal{Y}_{1,0,0}^{1/2}(x - z)^\dagger \psi(x) \right]^\mathsf{T} |3\rangle\langle 1| \left[ \int d^3y\, \mathcal{Y}_{1,0,0}^{1/2}(y - z)^\dagger \psi(y) \right], \tag{36}
\end{aligned}
$$

where the superscript $a$ in $S_{\text{inst}}^a$ serves to remind that this effective interaction is associated with a given adjoint mass $mt^a$, whose eigenvectors $|i\rangle$ feature in the vertex. However, at this point,

---

[5]This is the analog of the hermitian conjugate in Euclidean signature, and is discussed in Sec. 4.1.3.

we note that the role of the fermion mass is solely to regulate $K_{\pm} = (\det' \mathfrak{D}_{\pm}^{\dagger} \mathfrak{D}_{\pm})^{1/2}$ (our discussion below of reflection positivity will imply $K_+ = K_- \equiv K$), and the derived instanton-induced vertex is sensible in the massless limit, with the functional determinant being regulated in some other way. The adjoint mass then serves a role similar to a symmetry-breaking source for a specific ordered state in our calculation. When the massless limit, which does not "commute" with the resummation of the instanton gas, is taken at the end, the adjoint mass leaves behind in its wake a monopole which in turn will drive a confining transition into a proximate ordered state.

# 4 Monopole operators and their symmetries

We will now rewrite the 't Hooft vertex (36) using "zero-mode operators" in a form that makes explicit its relation to the CFT monopole operators constructed in Ref. [16]. To this end, we define the mode operators:

$$
\begin{aligned}
\bar{c}_{qjM}(z) &= \int d^3x \, \bar{\psi}(x) \mathcal{Y}_{qjM}^{j+1/2}(x-z), \\
c_{qjM}(z) &= \int d^3x \, \mathcal{Y}_{qjM}^{j+1/2}(x-z)^{\dagger} \psi(x), \\
c_{\pm 1,0,0} &\equiv d_{\pm},
\end{aligned}
\tag{37}
$$

where flavor indices have been suppressed. These can be thought of as a spacetime analog of a change of basis with coefficients $\langle jM|x\rangle$. In fact, this follows from a mode expansion of the fermion fields in monopole harmonics (see Eq. (7.3) of Ref. [29]), and thereby identifies $c_{qjM}$ as the $\mathbb{R}^3$ analog of the "zero-mode operators" of Refs. [16, 29], there defined in radial quantization on $S^2 \times \mathbb{R}$. The 't Hooft vertex (36) can be written in terms of these operators as:

$$
\begin{aligned}
S_{\text{inst}} = K \int d^3z \Big[ &e^{i\sigma(z)} \bar{d}_+(z) |2\rangle\langle 4| \bar{d}_+(z)^{\intercal} + e^{-i\sigma(z)} d_-(z)^{\intercal} |4\rangle\langle 2| d_-(z) \\
&+ e^{-i\sigma(z)} \bar{d}_-(z) |1\rangle\langle 3| \bar{d}_-(z)^{\intercal} + e^{i\sigma(z)} d_+(z)^{\intercal} |3\rangle\langle 1| d_+(z) \Big],
\end{aligned}
\tag{38}
$$

which should be understood as the monopole operator spawned by a given adjoint mass $mt^a$ with eigenvectors as in Eq. (20). This form makes it manifestly clear that the $\mathfrak{su}(4)$ part of the vertices, $|2\rangle\langle 4|$ and $|1\rangle\langle 3|$, must be antisymmetrized, in accordance with the observation in Refs. [16, 29] that monopole operators of minimal charge transform in the antisymmetric representation of the flavor group with $N_f/2$ indices. Before discussing flavor symmetry in greater detail, we first derive how the ZM operators (37) transform under spacetime symmetries, reflection positivity, and gauge transformations.

## 4.1 Spacetime symmetries, reflection positivity, and gauge invariance

### 4.1.1 Lorentz invariance

Since the ZM operator $\bar{d}_{\pm}$ defined in Eq. (37) creates a fermion in a $j=0$ state, one might intuitively expect it to be Lorentz invariant. To see that this bears out, consider a Lorentz transformation $\Lambda$ (rotation in Euclidean signature) with $U(\Lambda)$ the corresponding $SU(2)_{\text{rot}}$ action on spinors. Since the monopole spinor harmonics $\mathcal{Y}_{\pm 1,0,0}^{1/2}$ have total angular momentum

$j=0$, they must satisfy the identity:[6]

$$\mathcal{Y}^{1/2}_{\pm 1,0,0}(\Lambda x) = U(\Lambda)\mathcal{Y}^{1/2}_{\pm 1,0,0}(x), \tag{39}$$

using which we see that:

$$\begin{aligned}
\Lambda : \bar{d}_{\pm}(z) &\to \int d^3x\, \bar{\psi}(\Lambda^{-1}x)U^{\dagger}(\Lambda)\mathcal{Y}^{1/2}_{\pm 1,0,0}(x-z), \\
&= \int d^3x\, \bar{\psi}(x)U^{\dagger}(\Lambda)\mathcal{Y}^{1/2}_{\pm 1,0,0}\left(\Lambda(x-\Lambda^{-1}z)\right), \\
&= \int d^3x\, \bar{\psi}(x)\mathcal{Y}^{1/2}_{\pm 1,0,0}(x-\Lambda^{-1}z), \\
&= \bar{d}_{\pm}(\Lambda^{-1}z),
\end{aligned} \tag{40}$$

as expected of a Lorentz scalar.

### 4.1.2 $\mathcal{CRT}$

Here we consider how the ZM operators transform under the discrete symmetries of continuum Euclidean QED$_3$: reflection $\mathcal{R}$, charge conjugation $\mathcal{C}$, and time reversal $\mathcal{T}$. These are to be distinguished from the microscopic symmetries of the projective symmetry group (PSG) [4] for Dirac spin liquids on various lattices, to be discussed later in Sec. 5.

We define reflections $\mathcal{R}_\mu$ to be in the $\mu$-coordinate. Let us consider reflections $\mathcal{R}_1$ in the $x^1$ coordinate for concreteness. On spinors, this acts as $\psi \to \gamma_1 \psi$ and $\bar{\psi} \to \bar{\psi}(-\gamma_1)$ so that a flavor-singlet mass $\bar{\psi}\psi$ breaks reflection symmetry. Under $\mathcal{R}_1$, the unit vector $\hat{\varphi}=-(\sin\varphi)\hat{x}+(\cos\varphi)\hat{y} \to -\hat{\varphi}$ so that the monopole background in the Wu-Yang gauge [53] transforms as $\mathcal{A}_\mu=(0,0,\mathcal{A}_\varphi) \to (0,0,-\mathcal{A}_\varphi)$, which amounts to reversing the monopole charge $q \to -q$ as expected of reflections. Explicitly, the monopole spinor harmonics obey:

$$\mathcal{Y}^{1/2}_{\pm 1,0,0}(\theta,\pi-\varphi)=(-\gamma_1)\mathcal{Y}^{1/2}_{\mp 1,0,0}(\theta,\varphi), \tag{41}$$

under reflection $\mathcal{R}_1$ in the $x^1$-coordinate, so that

$$\begin{aligned}
\mathcal{R}_1 : \bar{d}_{\pm}(z) &\to \int d^3x\, \bar{\psi}(\mathcal{R}_1 x)(-\gamma_1)\mathcal{Y}^{1/2}_{\pm 1,0,0}(x-z), \\
&= \int d^3x\, \bar{\psi}(x)(-\gamma_1)\mathcal{Y}^{1/2}_{\pm 1,0,0}(\mathcal{R}_1(x-\mathcal{R}_1 z)), \\
&= \int d^3x\, \bar{\psi}(x)(-\gamma_1)^2\mathcal{Y}^{1/2}_{\mp 1,0,0}(x-\mathcal{R}_1 z), \\
&= \bar{d}_{\mp}(\mathcal{R}_1 z).
\end{aligned} \tag{42}$$

Charge conjugation is a unitary symmetry that acts to send:

$$\begin{aligned}
\psi &\to -\gamma_1\psi^* = -\gamma_1\gamma_3\bar{\psi}^{\mathsf{T}} = (i\gamma_2)\bar{\psi}^{\mathsf{T}}, \\
\bar{\psi} &\to \psi^{\mathsf{T}}(-\gamma_1\gamma_3) = \psi^{\mathsf{T}}(i\gamma_2),
\end{aligned} \tag{43}$$

---

[6]To explicitly verify this with the expressions for the harmonics in Appendix A requires some care, for such expressions are derived by solving the Euclidean Dirac equation in a fixed gauge. The background gauge field $\mathcal{A}_\mu dx^\mu$ also transforms under rotations, and one must make a subsequent gauge transformation to bring back $\mathcal{Y}^L_{qjM}$ to its original form, as discussed in Ref. [53].

which flips the sign of the Dirac current, but not the mass. Using:

$$(i\gamma_2)\mathcal{Y}^{1/2}_{\pm 1,0,0}(\theta,\varphi) = \pm\mathcal{Y}^{1/2}_{\mp 1,0,0}(\theta,\varphi)^*, \tag{44}$$

it is straightforward to verify that:

$$\mathcal{C}: \bar{d}_\pm(z) \to \pm d_\mp(z), \quad d_\pm(z) \to \mp\bar{d}_\mp(z). \tag{45}$$

In Euclidean signature, time reversal as a spacetime symmetry behaves identically to reflections [54], and is specifically unitary. It can be defined as:

$$\mathcal{T}: \psi(x) \to \gamma_3\psi(\mathcal{R}_3 x),$$
$$\bar{\psi}(x) \to \bar{\psi}(\mathcal{R}_3 x)\gamma_3, \tag{46}$$

where $\mathcal{R}_3$ is a reflection in the Euclidean time ($x^3$) coordinate. One can alternatively define a modified time-reversal operation $\mathcal{CT}$ that also involves charge conjugation. On the ZM operators,

$$\mathcal{T}: \bar{d}_\pm(z) \to \bar{d}_\mp(\mathcal{R}_3 z), \tag{47}$$

using the fact that, under $\mathcal{R}_3$ reflections,

$$\mathcal{Y}^{1/2,(N)}_{\pm 1,0,0}(\pi-\theta,\varphi) = \gamma_3\mathcal{Y}^{1/2,(S)}_{\mp 1,0,0}(\theta,\varphi), \tag{48}$$

as one can verify from explicit expressions for the monopole harmonics.

### 4.1.3 Reflection positivity

Reality of the real-time action (and thus unitarity of the corresponding quantum field theory) is guaranteed by *reflection positivity* $\vartheta(S)=S$ of the Euclidean action [51, 52]. This is a form of complex conjugation accompanied by a reversal of Euclidean time and an involution of the Grassmann algebra. With our choices of coordinates and Dirac matrices [24],

$$\vartheta(\lambda\psi(x)) := \lambda^*\bar{\psi}(\mathcal{R}_3 x)\gamma_3,$$
$$\vartheta(\lambda\bar{\psi}(x)) := \lambda^*\gamma_3\psi(\mathcal{R}_3 x), \quad \lambda\in\mathbb{C},$$
$$\vartheta(a_\mu(x)dx^\mu) := a_\mu(\mathcal{R}_3 x)d(\mathcal{R}_3 x)^\mu, \tag{49}$$

and $\vartheta$ also reverses the order of Grassmann variables, e.g., $\vartheta(\psi_\alpha\psi_\beta\psi_\gamma) = \vartheta(\psi_\gamma)\vartheta(\psi_\beta)\vartheta(\psi_\alpha)$. For instance, one can check that the usual Berry phase term $\int\bar{\psi}\gamma_3\partial_\tau\psi$ is reflection positive using the definitions above. On the ZM operators $d_\pm$, we observe that

$$\vartheta(\bar{d}_\pm(z)) = \int d^3x\,\mathcal{Y}^{1/2}_{\mp 1,0,0}(x-\mathcal{R}_3 z)^\dagger\psi(x),$$
$$= d_\mp(\mathcal{R}_3 z), \tag{50}$$

where we have used the fact that reflections invert the monopole charge [see Eq. (48)]. Together with the transformation $\vartheta(\sigma(z)) = \sigma(\mathcal{R}_3 z)$ for the dual photon [24], the transformation (50) ensures that the 't Hooft vertex (38) is reflection positive, thereby implying the reality of the real-time action or hermiticity of the Hamiltonian. It is important to note that reflection conjugation $\vartheta(d_q)$ replaces the notion of hermitian conjugation in Euclidean signature. In particular, we will *define*:

$$d^\dagger_\pm := \vartheta(d_\pm) = \bar{d}_\mp. \tag{51}$$

### 4.1.4 Local gauge invariance

We will prove invariance of the ZM operators under gauge transformations with nonzero support on a sphere of fixed radius in $\mathbb{R}^3$. By radial quantization, this suffices to prove gauge invariance in general. The integrand of the expressions (37) should be viewed as sections of a $U(1)$ bundle over punctured $\mathbb{R}^3$. Charting a fixed sphere surrounding the monopole with "northern" (N) and "southern" (S) gauges à la Wu-Yang [53], it is clear that $\psi$ should gauge transform identically to the spinor harmonics $\mathcal{Y}_{q00}^{1/2}$:

$$\psi^{(N)}(x) = e^{-iq\varphi}\psi^{(S)}(x),$$
$$\mathcal{Y}_{q,0,0}^{1/2,(N)}(x) = e^{-iq\varphi}\mathcal{Y}_{q,0,0}^{1/2,(S)}(x), \tag{52}$$

for $\varphi$ the azimuthal coordinate on $S^2 \subset \mathbb{R}^3$. Since $\int d^3x = \int_{N \cap S} d^3x$, because the N and S poles are a set of measure zero in the integral, the mode operators transform as:

$$\begin{aligned}
\bar{d}_q &= \int_{N \cap S} d^3x\, \bar{\psi}^{(N)}(x)\mathcal{Y}_{q,0,0}^{1/2,(N)}(x-z) \\
&= \int_{N \cap S} d^3x\, \bar{\psi}^{(S)}(x)(e^{-iq\varphi})^* e^{-iq\varphi}\mathcal{Y}_{q,0,0}^{1/2,(S)}(x-z) \\
&= \int_{N \cap S} d^3x\, \bar{\psi}^{(S)}(x)\mathcal{Y}_{q,0,0}^{1/2,(S)}(x-z) \\
&= \bar{d}_q.
\end{aligned} \tag{53}$$

A similar calculation shows invariance of $d_q$.

## 4.2 Flavor symmetry

The global form of the symmetry group of compact QED$_3$ with $N_f$ flavors has been nicely summarized in Ref. [55]; let us review the necessary aspects here in our framework and notation, for general $N_f$. The Lagrangian $\bar{\psi}\slashed{D}\psi$ is invariant under $U(N_f)$ rotations of the fermions, but the center $U(1)$ is a gauge redundancy as it leaves spin operators invariant. Moreover, it acts trivially on gauge-invariant fermion bilinears such as $\bar{\psi}t^a\psi$. One might then conclude that the symmetry group of the DSL is $PU(N_f) \times U(1)_\mathcal{M} \cong PSU(N_f) \times U(1)_\mathcal{M}$, where $PSU(N_f) \cong SU(N_f)/\mathbb{Z}_{N_f}$ and $U(1)_\mathcal{M}$ is the topological "magnetic" symmetry corresponding to conservation of magnetic charge $\frac{1}{2\pi}\int f$ on any 2-cycle. However, monopole operators do not transform well as a representation of this group. The monopoles of minimal charge are precisely the 't Hooft vertices calculated previously, and are of the form (for general $N_f$):

$$e^{i\sigma}\Delta_{a_1\cdots a_{N_f/2}}d_{a_1}^\dagger\cdots d_{a_{N_f/2}}^\dagger, \tag{54}$$

with $\Delta$ totally antisymmetric in its $N_f/2$ indices. Under the center of $SU(N_f)$ generated by $e^{2\pi i/N_f}$, the vertex transforms by an overall phase of $(e^{2\pi i/N_f})^{N_f/2} = -1$. This is identical to a $\pi$ shift in $U(1)_\mathcal{M}$, which implies the symmetry group is really:

$$\frac{SU(N_f) \times U(1)_\mathcal{M}}{\mathbb{Z}_{N_f}}, \tag{55}$$

where the $\mathbb{Z}_{N_f}$ in the quotient is generated by:

$$(e^{2\pi i/N_f}, -1) \in SU(N_f) \times U(1)_\mathcal{M}. \tag{56}$$

For $N_f = 4$, the isomorphism $SU(4)/\mathbb{Z}_2 \cong SO(6)$ can be used to equivalently write the symmetry group of the DSL as:

$$\frac{SO(6) \times U(1)_{\mathcal{M}}}{\mathbb{Z}_2}, \tag{57}$$

as concluded by Ref. [10].

A basis for the vector space of $q = \pm 1$ monopole operators can then be constructed from the six antisymmetric generators of $\mathfrak{su}(4)$. Doing so, we obtain three spin-singlet, valley-triplet monopoles:

$$
\begin{aligned}
e^{iq\sigma}\bar{d}_q(-i\sigma_2\mu_3)(\bar{d}_q)^{\intercal} &\equiv \mathcal{V}_{1q}, \\
e^{iq\sigma}\bar{d}_q(\sigma_2)(\bar{d}_q)^{\intercal} &\equiv \mathcal{V}_{2q}, \\
e^{iq\sigma}\bar{d}_q(i\sigma_2\mu_1)(\bar{d}_q)^{\intercal} &\equiv \mathcal{V}_{3q},
\end{aligned} \tag{58}
$$

and three spin-triplet, valley-singlet monopoles:

$$
\begin{aligned}
e^{iq\sigma}\bar{d}_q(-\sigma_3\mu_2)(\bar{d}_q)^{\intercal} &\equiv \mathcal{S}_{1q}, \\
e^{iq\sigma}\bar{d}_q(i\mu_2)(\bar{d}_q)^{\intercal} &\equiv \mathcal{S}_{2q}, \\
e^{iq\sigma}\bar{d}_q(\sigma_1\mu_2)(\bar{d}_q)^{\intercal} &\equiv \mathcal{S}_{3q}.
\end{aligned} \tag{59}
$$

It is straightforward to verify that these have the same spin/valley structure as the monopole operators defined in Refs. [9, 10], up to some signs chosen so that the six monopoles map to the standard basis of $\mathbb{C}^6$, under the isomorphism from the $\bigwedge^2 \mathbb{C}^4$ irrep of $SU(4)$ to the vector irrep of $SO(6)$. In addition, there are operators reflection conjugate to those defined above:

$$\mathcal{V}_{iq}^{\dagger} \equiv \vartheta(\mathcal{V}_{iq}), \qquad \mathcal{S}_{iq}^{\dagger} \equiv \vartheta(\mathcal{S}_{iq}), \tag{60}$$

which we can use to construct the six operators

$$\mathcal{V}_i = \mathcal{V}_{i+} + \mathcal{V}_{i-}^{\dagger}, \quad \mathcal{S}_i = \mathcal{S}_{i+} + \mathcal{S}_{i-}^{\dagger}, \tag{61}$$

For example,

$$\mathcal{V}_2 = e^{i\sigma}(\bar{d}_+\sigma_2\bar{d}_+^{\intercal} + d_+^{\intercal}\sigma_2 d_+), \tag{62}$$

is a monopole of definite magnetic charge $(+1)$ that can create or annihilate pairs of spinons, as illustrated earlier in Fig. 1.

By examining the instanton-induced 't Hooft vertex (38), we observe that a choice of $\mathfrak{su}(4)$-adjoint mass proliferates a linear combination of two of the six monopoles $\{\mathcal{V}_i, \mathcal{S}_i\}$. There are 15 such combinations, in correspondence with the 15 generators of $\mathfrak{su}(4)$. As an example, the 't Hooft vertex (38) for a spin-Hall mass $M_{30} = \bar{\psi}\sigma_3\psi$ can be written in the above basis as

$$
\begin{aligned}
\mathcal{L}_{30} &= \mathcal{S}_{1+} + i\mathcal{S}_{2+} + \mathcal{S}_{1-} - i\mathcal{S}_{2-} + \text{r.c.} \\
&= \operatorname{Re}\mathcal{S}_1 + \operatorname{Im}\mathcal{S}_2,
\end{aligned} \tag{63}
$$

defining $\operatorname{Re}\mathcal{S}_i \equiv \mathcal{S}_i + \mathcal{S}_i^{\dagger}$ and $\operatorname{Im}\mathcal{S}_i \equiv i(\mathcal{S}_i - \mathcal{S}_i^{\dagger})$. Again, the adjoint $\S_i^{\dagger}$ of a monopole operator should really be viewed in Euclidean signature as the "reflection conjugate" $\vartheta(\S_i)$ defined earlier in Sec. 4.1.3. In this way we can find the monopole operators spawned by all 15 adjoint masses, and we tabulate them in Table 1.

| Adjoint mass | Monopole proliferated |
|:---:|:---:|
| $M_{01}$ | $\mathcal{V}_3 + i\mathcal{V}_2 + \text{r.c.}$ |
| $M_{02}$ | $\mathcal{V}_3 + i\mathcal{V}_1 + \text{r.c.}$ |
| $M_{03}$ | $-\mathcal{V}_1 + i\mathcal{V}_2 + \text{r.c.}$ |
| $M_{i1}$ | $\mathcal{S}_i - i\mathcal{V}_1 + \text{r.c.}$ |
| $M_{i2}$ | $\mathcal{S}_i + i\mathcal{V}_2 + \text{r.c.}$ |
| $M_{i3}$ | $\mathcal{S}_i - i\mathcal{V}_3 + \text{r.c.}$ |
| $M_{i0}$ | $\mathcal{S}_j + i\mathcal{S}_k + \text{r.c.}$ |

Table 1: Monopoles proliferated by the 15 adjoint masses. "r.c." denotes the reflection conjugate. In the last row, $(ijk)$ is an even permutation of $(123)$.

## 5 Monopole quantum numbers on bipartite lattices

It was observed in Refs. [9, 10] that there exist orders on bipartite lattices whose microscopic symmetries are completely captured by appropriate adjoint masses. Using such orders, we can demand that the 't Hooft vertex induced by the given adjoint mass—i.e., the monopole proliferated by such a mass (Table 1)—must not break additional symmetries, in order to fix its quantum numbers under certain lattice symmetries. As we show below for the square lattice (Sec. 5.1) and the honeycomb lattice (Sec. 5.2), monopole quantum numbers on bipartite lattices are reproduced accurately by this method. We expect that this is true for any microscopic order that can be described in the continuum by condensing a fermion bilinear. Conversely, there exist conventional orders whose broken symmetries are not fully captured by condensing a fermion bilinear. Examples include the $\boldsymbol{q} = 0$ noncollinear magnetic states on the kagome lattice [8, 56]. Such orders have a $C_6$-breaking spin-ordering pattern which is invisible to all 15 adjoint masses, but is captured by the spin-triplet monopoles that serve as the correct order parameter for such states [9, 10]. (Precisely, it turns out that $C_6$ embeds into a $\mathbb{Z}_3^{\mathcal{M}}$ subgroup of $U(1)_{\mathcal{M}}$, as suspected initially in Ref. [8].) On non-bipartite lattices, monopole proliferation breaks additional symmetries beyond those broken by the adjoint mass [9], thus our method for determining monopole quantum numbers does not apply to those cases.

### 5.1 Square lattice

On a square lattice, a DSL is obtained by coupling a staggered flux mean-field state to $U(1)$ gauge fluctuations [5]. We work with the gauge choice of Refs. [9, 10] (but a different gamma matrix convention) which yields the following PSG action on the continuum Dirac spinor $\psi$:

$$
\begin{aligned}
T_x &: \psi \to (-i\sigma_2\mu_3)(i\gamma_2)\bar{\psi}^{\intercal}, & \bar{\psi} &\to \psi^{\intercal}(i\gamma_2)(i\sigma_2\mu_3), \\
T_y &: \psi \to (-i\sigma_2\mu_1)(i\gamma_2)\bar{\psi}^{\intercal}, & \bar{\psi} &\to \psi^{\intercal}(i\gamma_2)(i\sigma_2\mu_1), \\
r_x &: \psi \to (\mu_3\gamma_1)\psi, & \bar{\psi} &\to \bar{\psi}(-\gamma_1\mu_3), \\
C_{4s} &: \psi \to \frac{1}{\sqrt{2}}\sigma_2(i\mu_2-1)e^{-i\frac{\pi}{4}\gamma_2}(i\gamma_2)\bar{\psi}^{\intercal}, & \bar{\psi} &\to \psi^{\intercal}e^{-i\frac{\pi}{4}\gamma_2}(i\gamma_2)\sigma_2(-i\mu_2-1)\frac{1}{\sqrt{2}}, \\
\Theta &: \psi \to K i\mu_2(i\gamma_2)\gamma_3\bar{\psi}^{\intercal}, & \bar{\psi} &\to \psi^{\intercal}(-i\mu_2)(i\gamma_2)\gamma_3 K, & (64)
\end{aligned}
$$

for $x$ and $y$ translations ($T_x$, $T_y$), reflections in the $x$ coordinate ($r_x$), site-centered four-fold rotations ($C_{4s}$), and time reversal ($\Theta$), respectively, and $K$ denotes complex conjugation only on spin/valley matrices.

The embedding of the PSG into flavor (Sec. 4.2) and spacetime (Sec. 4.1) symmetries in the continuum completely fixes how the zero-mode part of the monopole operators transform. However, the lattice symmetries also embed into $U(1)_{\mathcal{M}}$, which acts on the bare monopole

|          | $T_x$ | $T_y$ | $r_x$ | $C_{4s}$  | $\Theta$ |
|----------|-------|-------|-------|-----------|----------|
| $M_{i0}$ | $-$   | $-$   | $-$   | $-$       | $-$      |
| $M_{01}$ | $-$   | $+$   | $+$   | $M_{03}$  | $+$      |
| $M_{03}$ | $+$   | $-$   | $-$   | $-M_{01}$ | $+$      |
| $M_{02}$ | $+$   | $+$   | $+$   | $-$       | $-$      |
| $M_{i1}$ | $+$   | $-$   | $+$   | $-M_{i3}$ | $+$      |
| $M_{i3}$ | $-$   | $+$   | $-$   | $M_{i1}$  | $+$      |
| $M_{i2}$ | $-$   | $-$   | $+$   | $+$       | $-$      |

Table 2: Transformation of the adjoint masses $M_{ij} = \bar{\psi}\sigma_i\mu_j\psi$ under the symmetries of the staggered-flux state on the square lattice.

$\exp(i\sigma)$, and this information is not present in the mean-field state from which the above PSG is derived. The most general approach to calculating this action, developed in Ref. [10], is to consider the Wannier limit, and the associated charge centers, of the spinon insulator obtained on gapping the DSL with a given adjoint mass. In this limit, the $U(1)_{\mathcal{M}}$ phase rotations of the monopole under lattice symmetries are interpreted as Aharonov-Bohm phases. For instance, a $C_{4s}$ action on a $q = +1$ monopole in an insulating state with gauge charges $Q$ at lattice sites will yield a phase $\exp(iQ\pi/2)$.

While no substitute for such rigorous microscopic arguments, we simply note here that the existence of orders whose symmetries are fully encapsulated by a fermion bilinear provides a simple means to compute some, if not all, of the monopole quantum numbers. For example, on the square lattice, the symmetries of Néel and valence-bond-solid (VBS) states are completely encapsulated in the adjoint masses $M_{i2}$ and $M_{01/3}$, respectively (see Table 2). Let us demand that the monopoles proliferated by those masses (Table 1) also remain invariant under the latter's symmetries. As the VBS mass $M_{03}$ is $T_x$ invariant, we require that the monopole $(-\operatorname{Re}\mathcal{V}_1 + \operatorname{Im}\mathcal{V}_2)$ also be $T_x$ invariant. Likewise, $C_{4s}$ is a symmetry of the Néel mass $M_{i2}$ which proliferates the monopole $\operatorname{Re}\mathcal{S}_i + \operatorname{Im}\mathcal{V}_2$. This means we can demand that $\operatorname{Im}\mathcal{V}_2 = i(\mathcal{V}_2 - \mathcal{V}_2^\dagger)$ be invariant under both $T_x$ and $C_{4s}$. However, from Eq. (64) we see the corresponding PSG transformations involve charge conjugation $\psi \to (i\gamma_2)\bar{\psi}^\intercal$. Thus, the ZM operators $d_\pm$ will also undergo charge conjugation [Eq. (45)], and from Eq. (58), $\mathcal{V}_2$ will be mapped to its reflection conjugate $\mathcal{V}_2^\dagger$. The only way for $\operatorname{Im}\mathcal{V}_2$ to remain invariant is thus to demand:

$$T_x(\mathcal{V}_2) = T_x(e^{i\sigma})(d_-^\intercal\sigma_2 d_- + \bar{d}_-\sigma_2\bar{d}_-^\intercal) \overset{!}{=} -\mathcal{V}_2^\dagger,$$

$$C_{4s}(\mathcal{V}_2) = C_{4s}(e^{i\sigma})(-d_-^\intercal\sigma_2 d_- - \bar{d}_-\sigma_2\bar{d}_-^\intercal) \overset{!}{=} -\mathcal{V}_2^\dagger, \tag{65}$$

which determines:

$$T_x(\sigma) = -\sigma + \pi, \qquad C_{4s}(\sigma) = -\sigma. \tag{66}$$

The quantum numbers of $\sigma$ under other lattice symmetries can be similarly calculated, but one can also exploit relational constraints among the generators of the PSG (see Supplemental Material of Ref. [9]). Using that $T_x T_y$ and $\Theta T_x$ are symmetries of the Néel order $M_{i2}$ leads to:

$$T_y(\sigma) = \Theta(\sigma) = -\sigma + \pi. \tag{67}$$

Finally, we look at reflections $r_x$ on the square lattice. Its embedding into the continuum symmetries involves the continuum reflection $\mathcal{R}_1$, which has an action $\mathcal{R}_1 \colon \bar{d}_\pm \to \bar{d}_\mp$ on ZM operators [Eq. (42)]. On the monopole $\mathcal{V}_2$, we find that:

$$r_x(\mathcal{V}_2) = r_x(e^{i\sigma})(\bar{d}_-\sigma_2\bar{d}_-^\intercal + d_-^\intercal\sigma_2 d_-) = e^{i\theta_r}\mathcal{V}_2^\dagger. \tag{68}$$

|  | $T_x$ | $T_y$ | $r_x$ | $C_{4s}$ | $\Theta$ |
|---|---|---|---|---|---|
| $\mathcal{V}_1$ | $\mathcal{V}_1^\dagger$ | $-\mathcal{V}_1^\dagger$ | $-\mathcal{V}_1^\dagger$ | $-\mathcal{V}_3^\dagger$ | $\mathcal{V}_1^\dagger$ |
| $\mathcal{V}_2$ | $-\mathcal{V}_2^\dagger$ | $-\mathcal{V}_2^\dagger$ | $-\mathcal{V}_2^\dagger$ | $-\mathcal{V}_2^\dagger$ | $-\mathcal{V}_2^\dagger$ |
| $\mathcal{V}_3$ | $-\mathcal{V}_3^\dagger$ | $\mathcal{V}_3^\dagger$ | $\mathcal{V}_3^\dagger$ | $\mathcal{V}_1^\dagger$ | $\mathcal{V}_3^\dagger$ |
| $\mathcal{S}_i$ | $-\mathcal{S}_i^\dagger$ | $-\mathcal{S}_i^\dagger$ | $\mathcal{S}_i^\dagger$ | $\mathcal{S}_i^\dagger$ | $-\mathcal{S}_i^\dagger$ |

Table 3: Monopole quantum numbers on the square lattice.

As reflections are a symmetry of the Néel mass $M_{i2}$, we can demand invariance under $r_x$ of the Im $\mathcal{V}_2$ monopole it proliferates. This sets $\theta_r = \pi$ in Eq. (68) and therefore

$$r_x(\sigma) = -\sigma + \pi. \tag{69}$$

The set of equations (66)-(69) completely determines the Berry phases of monopoles under the lattice symmetries (64). The total action of these symmetries on monopole operators has been summarized in Table 3. We note that our results are identical to the first four rows of Table 1 of Ref. [9]. In particular, we also find that the monopole Im $\mathcal{V}_2$ is trivial under all lattice symmetries.

We caution that one cannot expect the 't Hooft vertex to respect the symmetries of the adjoint mass *in general*, as demonstrated by the results of Refs. [9,10]. As an example, consider the unconventional order:

$$M_{i3} \sim \sum_r (-1)^{r_x} (\boldsymbol{S_r} \times \boldsymbol{S_{r+\hat{y}}})_i, \tag{70}$$

where the right-hand side is a spin operator on the square lattice with the same microscopic symmetries as the fermion bilinear on the left-hand side [5,9]. This equation suggests that $M_{i3}$ describes a spin-triplet VBS state invariant under $T_y$. However, the monopole proliferated by $M_{i3}$ is $\mathcal{S}_i - i\mathcal{V}_3$. By condensing this as $\langle \mathcal{S}_i - i\mathcal{V}_3 \rangle = 1 - i$, one observes that there is Néel order along $\sigma_i$ *in addition to* the order described by $M_{i3}$ (70). This follows from the fact that Re $\mathcal{S}_i$ and Re $\mathcal{V}_3$ have the symmetries of Néel order $M_{i2}$ and the triplet VBS order (70), respectively. The additional broken symmetries of the Néel order are not visible to the adjoint mass $M_{i3}$ but are captured by the associated 't Hooft vertex, which additionally breaks $T_y$ and $\Theta$ symmetries. However, the above method offers a quick way to compute quantum numbers when there exist orders with symmetries completely encoded in a fermion bilinear, paradigmatic examples being Néel and VBS orders.

## 5.2 Honeycomb lattice

On a honeycomb lattice, a parton mean-field Hamiltonian describing uniform nearest-neighbor hopping has a relativistic dispersion with gapless Dirac nodes at $K_\pm = \pm \frac{4\pi}{3\sqrt{3}}\hat{y}$. As is well-known, this model has a particle-hole symmetry which acts trivially on the physical spin operators, and when combined with $U(1)$ gauge fluctuations yields an $SU(2)$ gauge theory (QCD$_3$) at low energies [57]. However, the addition of longer-range hopping breaks particle-hole symmetry and yields a DSL described by CQED$_3$ in the infrared. Since the particle-hole symmetric state is adiabatically connected to the DSL, we may calculate monopole quantum numbers in the former for simplicity, and to make useful comparison with the results of Refs. [9, 10]. Choosing a two-site (AB) unit cell on armchair graphene with Bravais lattice

|            | $T_{1/2}$                      | $C_6$                           | $r_x$ | $\Theta$ |
|------------|--------------------------------|---------------------------------|-------|----------|
| $M_{i0}$   | $+$                            | $+$                             | $-$   | $+$      |
| $M_{01}$   | $\alpha M_{01}+\beta M_{02}$   | $\alpha M_{01}+\beta M_{02}$    | $+$   | $+$      |
| $M_{02}$   | $\alpha M_{02}-\beta M_{01}$   | $-\alpha M_{02}+\beta M_{01}$   | $-$   | $+$      |
| $M_{03}$   | $+$                            | $-$                             | $+$   | $+$      |
| $M_{i1}$   | $\alpha M_{i1}+\beta M_{i2}$   | $\alpha M_{i1}+\beta M_{i2}$    | $+$   | $-$      |
| $M_{i2}$   | $\alpha M_{i2}-\beta M_{i1}$   | $-\alpha M_{i2}+\beta M_{i1}$   | $-$   | $-$      |
| $M_{i3}$   | $+$                            | $-$                             | $+$   | $-$      |

Table 4: Transformation of the adjoint masses $M_{ij}=\bar{\psi}\sigma_i\mu_j\psi$ under the PSG (71) on the honeycomb lattice, with $\alpha=\cos\left(\frac{2\pi}{3}\right)$ and $\beta=\sin\left(\frac{2\pi}{3}\right)$.

vectors $\boldsymbol{a}_{1/2}=(1/2,\pm\sqrt{3}/2)$, the PSG for the particle-hole symmetric ansatz is:

$$
\begin{aligned}
T_{1/2}&: \psi \rightarrow e^{-i2\pi\mu_3/3}\psi, & \bar{\psi} &\rightarrow \bar{\psi}e^{i2\pi\mu_3/3}, \\
C_6&: \psi \rightarrow -i\mu_1 e^{-i2\pi\mu_3/3}e^{-i\pi\gamma_1/6}\psi, & \bar{\psi} &\rightarrow \bar{\psi}(ie^{i\pi\gamma_1/6}e^{i2\pi\mu_3/3}\mu_1), \\
r_x&: \psi \rightarrow \mu_2\gamma_3\psi, & \bar{\psi} &\rightarrow \bar{\psi}(-\mu_2\gamma_3) \\
\Theta&: \psi \rightarrow K(i\sigma_2\mu_2\gamma_3)\psi, & \bar{\psi} &\rightarrow \bar{\psi}(i\sigma_2\mu_2\gamma_3)K, 
\end{aligned}
\tag{71}
$$

for (respectively) translations $T_{1/2}$ along $\boldsymbol{a}_{1/2}$, plaquette-centered six-fold rotations ($C_6$), reflections about the vertical axis through an AB unit cell ($r_x$), and time reversal ($\Theta$). $K$ acts to complex conjugate only within the spin-valley space (i.e., the matrices $\sigma_i$ and $\mu_i$).

Similar to the square lattice in Sec. 5.1, we first tabulate the transformation of the 15 adjoint masses $M_{ij}=\bar{\psi}\sigma_i\mu_j\psi$ under the above PSG. From Table 4, it is clear that $M_{i3}$ encapsulates all the symmetries of Néel order on the honeycomb lattice. We can then expect the associated proliferated monopole $\mathcal{S}_i-i\mathcal{V}_3$ (see Table 1) to not break any additional symmetries, and thus demand:

$$
T_{1/2}(\mathcal{V}_3) = T_{1/2}(e^{i\sigma})[\bar{d}_+(i\sigma_2\mu_1)\bar{d}_+ - d_+(i\sigma_2\mu_1)d_+] \overset{!}{=} \mathcal{V}_3,
\tag{72}
$$

noting that to the difference of Eq. (64), the PSG here does not involve charge conjugation. Equation (72) implies that lattice translations act trivially on the dual photon. Turning to reflections, we similarly demand that $r_x(\mathcal{V}_3)=e^{i\theta_r}\mathcal{V}_3^\dagger$ be equal to $\mathcal{V}_3$, which leads to the action $r_x(\sigma)=-\sigma$ with no Berry phase.

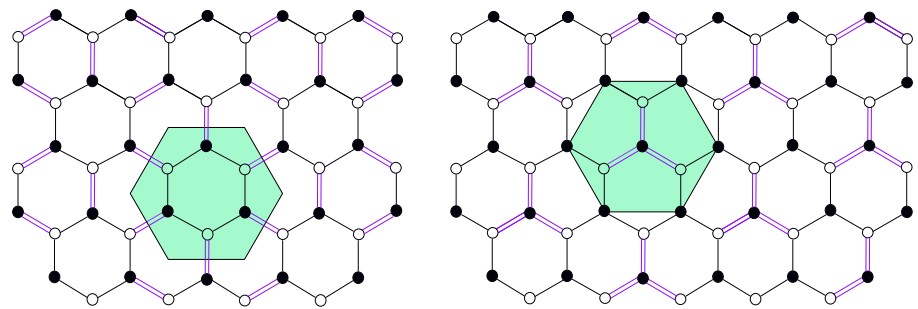

Figure 2: Kekulé-O (left) and Kekulé-Y (right) patterns on the honeycomb lattice.

Similarly, $M_{01}$ and $M_{02}$ account for all symmetries of the Kekulé-O and Kekulé-Y VBS states, respectively (Fig. 2). We can use the time-reversal invariance of these orders to demand invariance of the monopole $\text{Re}\,\mathcal{V}_3$ that both proliferate:

$$
\Theta(\mathcal{V}_3) = \Theta(e^{i\sigma})[\bar{d}_-(i\sigma_2\mu_1)\bar{d}_-^{\mathsf{T}} - d_-^{\mathsf{T}}(i\sigma_2\mu_1)d_-] = e^{i\theta_\Theta}\mathcal{V}_3^\dagger \overset{!}{=} \mathcal{V}_3^\dagger,
\tag{73}
$$

and so $\theta_\Theta = 0$. Finally, to compute quantum numbers under $C_6$, we may use the $C_6\Theta$ symmetry of the Néel mass $M_{i3}$ to demand invariance of $\text{Im}\,\mathcal{V}_3$:

$$
\begin{aligned}
(C_6 \circ \Theta)(i(\mathcal{V}_3 - \mathcal{V}_3^\dagger)) &= -iC_6(\mathcal{V}_3 - \mathcal{V}_3^\dagger), \\
&= -ie^{-i\theta_6}(\mathcal{V}_3 - \mathcal{V}_3^\dagger), \\
&\overset{!}{=} i(\mathcal{V}_3 - \mathcal{V}_3^\dagger),
\end{aligned}
\tag{74}
$$

which requires $\theta_6 = \pi$. Collecting our results, the lattice symmetries act on the dual photon as follows:

$$
\begin{aligned}
T_{1/2}(\sigma) &= \sigma, \\
r_x(\sigma) &= -\sigma, \\
\Theta(\sigma) &= -\sigma, \\
C_6(\sigma) &= \sigma + \pi,
\end{aligned}
\tag{75}
$$

from which transformations of all six monopole operators can be determined. These results are summarized in Table 5, and agree with the results in Table 1 of Ref. [9] for the honeycomb lattice. The monopole $\text{Re}\,\mathcal{V}_3$ is trivial under all lattice symmetries and is thus a symmetry-allowed perturbation to the DSL.

|  | $T_{1/2}$ | $r_x$ | $C_6$ | $\Theta$ |
|---|---|---|---|---|
| $\mathcal{V}_1$ | $\alpha\mathcal{V}_1 - \beta\mathcal{V}_2$ | $\mathcal{V}_1^\dagger$ | $-\alpha\mathcal{V}_1 + \beta\mathcal{V}_2$ | $\mathcal{V}_1^\dagger$ |
| $\mathcal{V}_2$ | $\alpha\mathcal{V}_2 + \beta\mathcal{V}_1$ | $-\mathcal{V}_2^\dagger$ | $\alpha\mathcal{V}_2 + \beta\mathcal{V}_1$ | $\mathcal{V}_2^\dagger$ |
| $\mathcal{V}_3$ | $\mathcal{V}_3$ | $\mathcal{V}_3^\dagger$ | $\mathcal{V}_3$ | $\mathcal{V}_3^\dagger$ |
| $\mathcal{S}_i$ | $\mathcal{S}_i$ | $-\mathcal{S}_i^\dagger$ | $-\mathcal{S}_i^\dagger$ | $-\mathcal{S}_i^\dagger$ |

Table 5: Monopole quantum numbers on the honeycomb lattice, with $\alpha = \cos\left(\frac{2\pi}{3}\right)$ and $\beta = \sin\left(\frac{2\pi}{3}\right)$.

# 6  Conclusion

In summary, we have constructed monopole operators in the DSL directly on $\mathbb{R}^3$ without assuming conformal invariance, and have computed their quantum numbers under lattice symmetries on the square and honeycomb (bipartite) lattices. The first task was accomplished by first deforming the DSL with a choice of an $\mathfrak{su}(4)$-valued fermion mass. This was shown to lead to ZMs of the Euclidean Dirac operator exponentially bound to monopole-instantons. The interpretation of these ZMs in the Hamiltonian framework and their relation to zero-energy modes was also discussed. We then showed that resumming a semiclassical instanton gas in the presence of such ZMs leads to an instanton-induced effective interaction, designated as the 't Hooft vertex in analogy with a similar effect in QCD$_4$. By introducing ZM creation/annihilation operators, we then identified this vertex as a linear combination of two of six possible monopole operators in the DSL, previously constructed in radially-quantized conformal CQED$_3$.

Our next result involved an analysis of the effects of lattice symmetries in specific microscopic realizations of the DSL. By recognizing the existence of orders on bipartite lattices with symmetries fully encapsulated in a specific fermion bilinear, we were able to compute quantum numbers of all monopoles under symmetries of the DSL on square and honeycomb lattices. Specifically, from a symmetry standpoint, Néel and VBS orders on these lattices could

be described in the continuum by *either* appropriate fermion bilinears *or* monopole operators (although monopole proliferation is necessary to confine spinons). By knowing the 't Hooft vertex associated to a given bilinear, we could then demand that the former *not* break additional symmetries of the DSL to fix the lattice symmetry action on monopoles. Néel and VBS orders on the square and honeycomb lattices together possess enough unbroken lattice symmetries to fully determine the transformations of all monopole operators. In particular, our results for the "Berry phase" of monopoles, arising from the embedding of the lattice symmetries into the magnetic symmetry $U(1)_{\mathcal{M}}$ of the dual photon, were shown to be consistent with the more general Wannier center calculations of Refs. [9, 10]. On both square and honeycomb lattices, we showed the existence of a monopole transforming trivially under all lattice symmetries, and thus an allowed perturbation to the DSL on these lattices likely to lead to its instability.

# Acknowledgements

We thank S. Dey for useful discussions. G.S. was supported by the Golden Bell Jar Graduate Scholarship in Physics. J.M. was supported by NSERC Discovery Grants RGPIN-2020-06999 and RGPAS-2020-00064; the Canada Research Chair (CRC) Program; the Government of Alberta's Major Innovation Fund (MIF); and the Pacific Institute for the Mathematical Sciences (PIMS) Collaborative Research Group program.

# A  Zero modes of Euclidean Dirac operators

Since an index theorem for Dirac operators with abelian gauge fields on odd-dimensional noncompact manifolds has not been established, we resort to an explicit calculation of zero modes.

A charge-$q \in \mathbb{Z}$ monopole-instanton can be described by a Wu-Yang connection,

$$\mathcal{A}_q = \begin{cases} -\frac{q}{2r\sin\theta}(\cos\theta - 1)\hat{\boldsymbol{\varphi}}, & \theta \in (0, \pi/2), \\ -\frac{q}{2r\sin\theta}(\cos\theta + 1)\hat{\boldsymbol{\varphi}}, & \theta \in (\pi/2, \pi), \end{cases} \tag{76}$$

in spherical coordinates with an orthonormal frame $(\hat{\boldsymbol{r}}, \hat{\boldsymbol{\theta}}, \hat{\boldsymbol{\varphi}})$. We will explicitly solve for the zero modes of the Euclidean (non-self-adjoint) Dirac operator in an instanton background,

$$\mathfrak{D}_{aq} = \slashed{\partial} - i\mathcal{A}_q + mt^a, \tag{77}$$

where $t^a \in \mathfrak{su}(4)$. Using the fact that $(\boldsymbol{\gamma} \cdot \hat{\boldsymbol{r}})^2 \equiv \gamma_r^2 = 1$, with $\boldsymbol{\gamma} = (\gamma_1, \gamma_2, \gamma_3)$ the Pauli vector, the Dirac operator can be rewritten as [16, 24]:

$$\gamma_r^2 \mathfrak{D}_{aq} = \gamma_r \left( \partial_r - \frac{1}{r}\boldsymbol{\gamma} \cdot \boldsymbol{L} - \frac{q}{2r}\gamma_r \right) + mt^a, \tag{78}$$

where:

$$\boldsymbol{L} = \boldsymbol{r} \times (\boldsymbol{p} - \boldsymbol{a}) - \frac{q}{2}\hat{\boldsymbol{r}}, \tag{79}$$

is the conserved angular momentum in a monopole field. Defining the total angular momentum:

$$\boldsymbol{J} = \boldsymbol{L} + \frac{1}{2}\boldsymbol{\gamma}, \tag{80}$$

the Dirac operator takes the form:

$$\mathfrak{D}_{aq} = \gamma_r \left[ \partial_r - \frac{1}{r}(\boldsymbol{J}^2 - \boldsymbol{L}^2 - \frac{3}{4}) - \frac{q}{2r}\gamma_r \right] + mt^a. \tag{81}$$

To find the eigenfunctions, note that $\boldsymbol{J}^2$, $J_z$, $t^a$ and $\mathfrak{D}_{aq}$ commute. This prompts an eigenfunction ansatz:

$$u^{qi}_{jM} = R(r)\mathcal{Y}^{j+1/2}_{qjM}(\theta, \varphi)|i\rangle_a + S(r)\mathcal{Y}^{j-1/2}_{qjM}(\theta, \varphi)|i\rangle_a, \tag{82}$$

where $|i\rangle_a$ is one of the four eigenvectors of $t^a$ with eigenvalue $(-1)^i$, and $\mathcal{Y}^L_{qjM}$ are monopole spinor harmonics defined in Appendix A of Ref. [24], and also in [16]. Their necessary properties are summarized as follows:

$$\begin{aligned}
\boldsymbol{J}^2 \mathcal{Y}^L_{qjM} &= j(j+1)\mathcal{Y}^L_{qjM}, \\
\boldsymbol{L}^2 \mathcal{Y}^L_{qjM} &= L(L+1)\mathcal{Y}^L_{qjM}, \\
J_z \mathcal{Y}^L_{qjM} &= M\mathcal{Y}^L_{qjM}, \\
\gamma_r \mathcal{Y}^{j\pm1/2}_{qjM} &= a_\pm \mathcal{Y}^{j+1/2}_{qjM} + b_\pm \mathcal{Y}^{j-1/2}_{qjM},
\end{aligned} \tag{83}$$

where:

$$\begin{aligned}
j &\in \left\{ \frac{|q|}{2} - \frac{1}{2}, \frac{|q|}{2} + \frac{1}{2}, \dots \right\}, \ (j > 0), \\
M &\in \{-j, -j+1, \dots, j\}, \\
L &\in \left\{ j - \frac{1}{2}, j + \frac{1}{2} \right\}, \ \left( L \geq \frac{|q|}{2} \right), \\
a_+ &= -b_- = \frac{q}{2j+1}, \quad a_- = b_+ = -\frac{\sqrt{(2j+1)^2 - q^2}}{2j+1}.
\end{aligned} \tag{84}$$

The condition $j > 0$ implies $j = (|q|-1)/2$ is excluded when $q = 0$, and the condition $L \geq |q|$ requires that $L = j - 1/2$ be excluded when $j = (|q|-1)/2$. Therefore, for a fixed $q$, the lowest angular momentum states with $j = (|q|-1)/2$ have $S(r) = 0$ in the ansatz (82). As we shall now show, these states are zero modes. The zero mode equation for $\mathfrak{D}_{aq}$ then separates to:

$$\left( \partial_r R + \frac{1}{r}R + \text{sgn}(q)m_i \right) R(r) = 0, \tag{85}$$

where $m_i = (-1)^i m$ corresponding to $|i\rangle_a$, the $SU(4)$ part of the zero mode. Solving for the radial function $R(r)$, the zero modes can be written as:

$$\begin{aligned}
u^{qi}_{(q-1)/2,M} &= R^{qi}(r)\mathcal{Y}^q_{q,(q-1)/2,M}(\theta, \varphi)|i\rangle_a, \\
&= \frac{\sqrt{2m}}{r} e^{-\text{sgn}(q)(-1)^i mr} \mathcal{Y}^q_{q,(q-1)/2,M}|i\rangle_a.
\end{aligned} \tag{86}$$

For a fixed monopole charge $q$ and $\mathfrak{su}(4)$ mass $mt^a$, it is clear that there are $2q \times \frac{N_f}{2} = qN_f$ linearly independent *normalizable* zero modes.[7] We have utilized the fact that for $j = (q-1)/2$, the quantum number $M$ ranges over the $2q$ values $-j, \dots, j$, and that a given sign of $q$ results in precisely two of the four eigenvectors $|i = 1, 2, 3, 4\rangle_a$ contributing normalizable ZMs.

---

[7]It is assumed that $N_f$ is even, so that there is no parity anomaly. In the case of $N_f$ odd, the non-anomalous theory has a half-integral Chern-Simons term that can be regarded as the result of integrating out an extra Dirac fermion.

It is also important to consider zero modes of the adjoint Dirac operator $\mathfrak{D}^\dagger$, for the Dirac action can be rewritten after an integration by parts and throwing away boundary terms as:

$$
\begin{aligned}
S_f &= \int d^3x\, \bar{\psi}(\slashed{\partial} - i\slashed{a} + mt^a)\psi, \\
&= \int d^3x\, [(-\slashed{\partial} + i\slashed{a} + mt^a)\bar{\psi}^\dagger]^\dagger \psi,
\end{aligned}
\tag{87}
$$

where it is to be remembered that $\bar{\psi}$ and $\psi$ are independent variables in the Euclidean path integral, unrelated by any notion of complex conjugation. Repeating the calculation above leads to the zero modes:

$$
\begin{aligned}
v^{qi}_{(q-1)/2,M} &= R^{qi}(r)\mathcal{Y}^q_{q,(q-1)/2,M}(\theta,\varphi)|i\rangle_a, \\
&= \frac{\sqrt{2m}}{r}e^{\mathrm{sgn}(q)(-1)^i mr}\mathcal{Y}^q_{q,(q-1)/2,M}|i\rangle_a,
\end{aligned}
\tag{88}
$$

where again, for a given $q$, $i$ must be chosen to ensure normalizability.

For reference, we give expressions for the monopole spinor harmonics, also given in Appendix A of Ref. [24]:

$$
\begin{aligned}
\mathcal{Y}^{j-1/2}_{q,j,m}(\theta,\varphi) &= \frac{1}{\sqrt{2j}}\begin{pmatrix} \sqrt{j+m_j}\, Y_{q,j-\frac{1}{2},m-\frac{1}{2}} \\ \sqrt{j-m}\, Y_{q,j-\frac{1}{2},m+\frac{1}{2}} \end{pmatrix}, \\
\mathcal{Y}^{j+1/2}_{q,j,m}(\theta,\varphi) &= \frac{1}{\sqrt{2j+2}}\begin{pmatrix} -\sqrt{j-m_j+1}\, Y_{q,j+\frac{1}{2},m-\frac{1}{2}} \\ \sqrt{j+m+1}\, Y_{q,j+\frac{1}{2},m+\frac{1}{2}} \end{pmatrix}.
\end{aligned}
\tag{89}
$$

The monopole harmonics $Y_{qLM}$ are defined in terms of the Wigner $D$-matrices $D^J_{MM'}(\alpha,\beta,\gamma)$ [53,58]. In the northern chart on a sphere that surrounds the monopole-instanton,

$$
Y_{q,L,M}(\theta_N,\varphi) = \sqrt{\frac{2L+1}{4\pi}}\left[D^L_{M,-q/2}(\varphi,\theta,-\varphi)\right]^*,
\tag{90}
$$

where $\theta_N \in [0,\pi)$. The southern versions (which are valid on the south pole) can be obtained via a gauge transformation on the overlapping region between northern and southern charts:

$$
Y_{q,L,M}(\theta_S,\varphi) = e^{-i2q\varphi}Y_{q,L,M}(\theta_N,\varphi).
\tag{91}
$$

From the above formula, the first two $q=1$ harmonics are given by

$$
\begin{aligned}
Y_{1,\frac{1}{2},\frac{1}{2}}(\theta_N,\varphi) &= -\frac{1}{\sqrt{2\pi}}e^{i\varphi}\sin\frac{\theta}{2}, \\
Y_{1,\frac{1}{2},-\frac{1}{2}}(\theta_N,\varphi) &= \frac{1}{\sqrt{2\pi}}\cos\frac{\theta}{2},
\end{aligned}
\tag{92}
$$

in the northern chart. Their analogs on the southern chart are obtained from the gauge transformation $\exp(-i\varphi)$.

For $q=-1$, the first two harmonics on the northern chart are

$$
\begin{aligned}
Y_{-1,\frac{1}{2},\frac{1}{2}}(\theta_N,\varphi) &= \frac{1}{\sqrt{2\pi}}\cos\frac{\theta}{2}, \\
Y_{-1,\frac{1}{2},-\frac{1}{2}}(\theta_N,\varphi) &= \frac{1}{\sqrt{2\pi}}e^{-i\varphi}\sin\frac{\theta}{2},
\end{aligned}
\tag{93}
$$

with their versions in the southern chart now obtained from the gauge transformation $\exp(i\varphi)$.

## B  Real-space Dirac propagator

With the Lagrangian $\bar{\psi}(\not{\partial}+m)\psi$ where $m$ is a signed quantity, the free Dirac propagator on $\mathbb{R}^3$ is:

$$
\begin{aligned}
G_f(x) &= \int \frac{d^3k}{(2\pi)^3} e^{ikx} \frac{i\not{k}-m}{k^2+m^2}, \\
&= (\not{\partial}-m)\int \frac{d^3k}{(2\pi)^3} \frac{e^{ikx}}{k^2+m^2}, \\
&= (\gamma_r \partial_r - m)\frac{e^{-|m|r}}{4\pi r}, \\
&= -\frac{e^{-|m|r}(1+|m|r)}{4\pi r^2}\gamma_r - \frac{me^{-|m|r}}{4\pi r}.
\end{aligned}
\tag{94}
$$

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
