# Peer review of "Monopoles in Dirac spin liquids and their symmetries from instanton calculus"

_SciPost Physics_

## Round 1 · Referee Report · Anonymous (Referee 1) · 2023-12-27

Strengths

see report

Weaknesses

see report

Report

This paper uses familiar semiclassical field theory methods to study instanton effects in $N_f = 4$ U(1) Dirac spin liquids. It does two things: it derives the form of the nonvanishing monopole operators in the presence of a certain SU(4)-adjoint fermion mass perturbation (previously derived using the state-operator correspondence), and it determines the quantum numbers of monopole operators on honeycomb and square lattices (previously derived by Song et al using microscopic tricks involving Wannier centers). While not a big surprise, the paper is clear and valuable should be published.

Small comments:

-- first paragrah, page 2: "glued together by a confining gauge field" The adjective "confining" is extremely confusing and misleading here, since as the next sentence correctly points out, it's only when the gauge field is deconfined that this construction describes a spin liquid.

-- bottom of page 2: "on realizations of the DSL..." -> "in realizations of the DSL"

-- bottom of page 3: I found the phrase "kill the path integral" unpleasant and obscure.

-- after equation (3) the $f$s in the Nambu spinor should be $c$s.

-- page 5 before equation (10): "On the other hand, a trivial $SU(2)$ flux..." perhaps should be replaced by something like "In contrast, a trivial $SU(2)$ flux..." to emphasize better that this option is not compatible with the situation described in the previous sentence.

-- Right before equation (10), I didn't understand in what sense the fields $\psi_{\alpha\sigma}$ are accounting for $U(1)$ gauge fluctuations -- aren't these the fermions??

-- Why not use the standard notation $ \sigma_x, \sigma_y, \sigma_z$ for the Pauli matrices, rather than $\gamma...$ ?

-- first paragraph of section 3: I found the sentence "In contrast to the massless case, where no such Euclidean ZMs exist" to be unnecessarily obscure. I suppose the authors are saying that as $m \to 0$, the modes become non-normalizible?

-- The discussion surrounding equation (13) regarding the dilute instanton gas approximation is unnecessarily formal and confusing. Specifically I really didn't understand the picture the authors describe with "fields localized in large boxes". The fields $\psi$ are presumably supported everywhere in spacetime. Is the idea that they are somehow expanding the fields in a basis of modes which are like a partition of unity? This seems unnecessary to me.

-- The caption to Figure 1 needs to be improved to explain each part a - d. I see that they are each referenced in the text, but it is not exactly clear what is happening in each of these pictures.

-- Two things about the sentence: "Of course, an instanton is the paradigm of a non-adiabatic process." 1. Although dictionaries agree with this usage of the word 'paradigm' to mean 'archetype', I found it confusing. I guess this is because: "The scientific community has added to the confusion by using it to mean "a theoretical framework," a sense popularized by American scientist Thomas S. Kuhn in the second edition of his influential book The Structure of Scientific Revolutions, published in 1970. In that work, Kuhn admitted that he had used paradigm in 22 different ways. Some usage commentators now advise avoiding the term entirely on the grounds that it is overused." 2. Whatever word is used, it is not at all clear to me that the sentence is true. At least in UV complete field theories, instantons are smooth, slowly-varying field configurations, where adiabatic methods can be quite useful.

-- Probably a little more detail can be included in the definition of $\Delta_\pm$. What is the meaning of "vertex factor"?

-- I get the impression that the mass term the authors added is not the most general possibility. Can the authors say something about the instantons for other choices of symmetry-breaking pattern (perhaps not in as much detail)?

-- A question of presentation: a naive reader might get the wrong idea that these methods have not previously been used except by 't Hooft, and not previously been used for condensed matter problems, neither of which is at all the case.

Requested changes

see report

  • validity: top
  • significance: -
  • originality: -
  • clarity: -
  • formatting: -
  • grammar: -

Author:  G. Shankar  on 2024-03-11  [id 4357]

(in reply to Report 1 on 2023-12-27)
Category:
answer to question

We thank the referee for comments on our work and their recommendation to publish. We also thank the referee for the list of typos and suggested grammatical corrections, which have been implemented in the revised manuscript. Below we respond to questions raised in their report.

Right before equation (10), I didn’t understand in what sense the fields $\psi_{\alpha\sigma}$ are accounting for $U(1)$ gauge fluctuations -- aren’t these the fermions?

Indeed, it is instead the gauge field $a_\mu$ that accounts for $U(1)$ gauge fluctuations. The phrasing has been changed in the revised manuscript to “A linearized description at these nodes with low-energy fermions $\psi_{\alpha\sigma}$, that also accounts for $U(1)$ gauge fluctuations with an emergent gauge field $a_\mu$, is then given by the Lagrangian. . . "

Why not use the standard notation $\sigma_x,\sigma_y,\sigma_z$ for the Pauli matrices, rather than $\gamma$...?

There are three sets of Pauli matrices used in the paper, which act on valley/nodal, physical spin, and Dirac indices of the fermion $\psi$. We use $\sigma_i$ to denote the matrices that act within the spin subspace, $\mu_i$ for the matrices within the nodal subspace, and $\gamma_i$ for the Dirac matrices, in order to avoid confusion in expressions such as equation (64).

first paragraph of section 3: I found the sentence "In contrast to the massless case, where no such Euclidean ZMs exist" to be unnecessarily obscure. I suppose the authors are saying that as $m\to 0$ , the modes become non-normalizible?

Indeed, the zero modes found become non-normalizable in the massless limit $m\to 0$. We have edited the quoted sentence to better reflect this fact.

The discussion surrounding equation (13) regarding the dilute instanton gas approximation is unnecessarily formal and confusing. Specifically I really didn’t understand the picture the authors describe with "fields localized in large boxes". The fields $\psi$ are presumably supported everywhere in spacetime. Is the idea that they are somehow expanding the fields in a basis of modes which are like a partition of unity? This seems unnecessary to me.

This is an argument introduced by ’t Hooft in Refs. [43,44], and is just the standard dilute gas approximation extended to the case where fermions are also present. Suppose we are given a multi-instanton configuration – for example, two monopoles at locations $z_1$ and $z_2$. If $z_1$ and $z_2$ are far apart (dilute gas assumption), then the fermion ($\psi$) profile can be written as a sum of two pieces, each with support only in a small neighbourhood of $z_1$ or $z_2$. This allows the fermion path integral to brought inside the instanton gas sum in equation (13).

The caption to Figure 1 needs to be improved to explain each part a - d. I see that they are each referenced in the text, but it is not exactly clear what is happening in each of these pictures.

We have extended the caption to the quoted figure in the revised version to elucidate the meaning of the subfigures.

Two things about the sentence: "Of course, an instanton is the paradigm of a non-adiabatic process." 1. Although dictionaries agree with this usage of the word ’paradigm’ to mean ’archetype’, I found it confusing. I guess this is because: "The scientific community has added to the confusion by using it to mean "a theoretical framework," a sense popularized by American scientist Thomas S. Kuhn in the second edition of his influential book The Structure of Scientific Revolutions, published in 1970. In that work, Kuhn admitted that he had used paradigm in 22 different ways. Some usage commentators now advise avoiding the term entirely on the grounds that it is overused." 2. Whatever word is used, it is not at all clear to me that the sentence is true. At least in UV complete field theories, instantons are smooth, slowly-varying field configurations, where adiabatic methods can be quite useful.

The purport of the quoted sentence was to simply point out that instantons are localized in space and time, but we agree that it is a confusing statement in light of the adiabatic arguments presented in that section, and have edited it out of the revised manuscript.

Probably a little more detail can be included in the definition of $\Delta_\pm$. What is the meaning of "vertex factor”?

The calculation of $\Delta_\pm$ is carried out in detail in section 3.3. For instance, these $su(4)$ matrices can be read off from equation (38). Earlier in section 3, they are only heuristically introduced, on general grounds as $su(4)$ matrices that act as a flavor rotation of the fermion $\psi$.

In QCD4, the instanton-induced interaction term is called a ’t Hooft vertex in the high-energy literature. We have borrowed the term for the analogous instanton-induced term in compact QED3 that appears in our paper. The term “vertex factor” is heuristically borrowed from perturbation theory. For instance, the standard QED (photon-fermion) vertex factor is $e\gamma_\mu$. By analogy, the interaction of a monopole with a pair of fermions has a vertex factor $\Delta_\pm$.

I get the impression that the mass term the authors added is not the most general possibility. Can the authors say something about the instantons for other choices of symmetry-breaking pattern (perhaps not in as much detail)?

The 15 mass terms considered [equation (11)] are the most general Hermitian and Lorentz- invariant fermion masses that do not radiatively generate a Chern-Simons term for the gauge field $a_\mu$. Indeed, along with the identity matrix, they form a basis for the space of all 4×4 Hermitian matrices. The identity itself, that is the fermion mass $\bar{\psi}\psi$, generates a Chern-Simons term for $a_\mu$ at one-loop and is not expected to lead to a symmetry-broken phase with confined gauge charges. This is because the bare monopole operator is no longer gauge invariant in the presence of a CS term and thus cannot be proliferated. We have added a comment on this point in the revised manuscript.

A question of presentation: a naive reader might get the wrong idea that these methods have not previously been used except by ’t Hooft, and not previously been used for condensed matter problems, neither of which is at all the case.

While instanton methods are of course not novel in either high-energy or condensed matter physics, their use in gauge theories with matter, particularly the effects of instanton-bound fermion zero modes, has to the best of our knowledge little precedent in the condensed matter literature. Most existing works (e.g. Refs [9,10,16-22]) have rather focused on conformal (state-operator) methods to construct monopole operators. Again to the best of our knowledge, in the high-energy literature, such methods have been used (after ’t Hooft) most notably by M. Ünsal in various studies of confinement in 3D and 4D gauge theories with matter. We have added a citation [40] to a related paper in the revised version to provide better context for the methods used. We shall also be grateful if any omissions in our citations to related work is brought to our attention.

---

## Round 1 · Referee Report · Anonymous (Referee 2) · 2024-2-14

Strengths

In report.

Weaknesses

In report.

Report

The paper studies monopole operators in Dirac spin liquids, with a path-integral instanton approach. The main results have to do with monopole proliferation due to various fermion mass-terms. The approach pertains to DSLs on bi-partite lattices, as explained by the authors. The final results pertain to the quantum numbers of selected monopoles, and were obtained in with mean-field-like methods. To the best of my knowledge, these results are robust, which is further confirmed by their agreement with numerous previous works. Although technical, the paper is well-written, and I applaud the authors for including physical intuition at various points, as well as a clear description of lattice considerations. I recommend publication once the authors address the following comment:

I believe eq.13 requires further justification. Is the dilute gas approximation justified by a large-Nf expansion or large-mass expansion? Otherwise the theory is not tractable. As this is the underlying method of the work, it certainly deserves a clearer explanation.

And these small points:
- Below eq.3, f operators not defined. I’m assuming c was meant.

  • “is now independent under an SU(2)” : I’m assuming “invariant” was meant

Requested changes

In report

  • validity: high
  • significance: ok
  • originality: good
  • clarity: high
  • formatting: excellent
  • grammar: perfect

Author:  G. Shankar  on 2024-03-11  [id 4358]

(in reply to Report 2 on 2024-02-14)
Category:
answer to question

We thank the referee for their comments on our work and recommendation to publish. The typos noted in the report have been corrected in the revised version.

We thank the referee for the pertinent question on the validity of the dilute gas approximation. In the pure gauge theory (without matter), monopoles interact via a Coulomb potential, which is known not to form bound states in 3D. The diluteness of this Coulomb gas can be argued using a standard Debye screening theory that assumes weak coupling $e^2 \ell\!\ll\!1$ (where $\ell$ is a UV cutoff), as in Polyakov’s original work (Refs. [13-15]). In the theory with fermion matter considered in this work, there exist pair exchanges of fermions between monopoles as shown in the paper. Such pair exchanges can be expected to correct the Coulomb potential between bare monopoles. However, since the fermions considered in the paper are massive, we suspect that such a correction is not enough to overwhelm the long-range Coulomb interaction and result in monopole bound states that would invalidate a dilute gas approximation. This discussion has been added to the revised version below equation (13).

---

## Editorial Decision

resubmitted